# Historical trends and controlling factors of isoprene emissions in CMIP6 Earth system models TS1

**Ngoc Thi Nhu Do**[1], **Kengo Sudo**[1,3], **Akihiko Ito**[2,3,4], **Louisa K. Emmons**[5], **Vaishali Naik**[6], **Kostas Tsigaridis**[7,8], **Øyvind Seland**[9], **Gerd A. Folberth**[10], **and Douglas I. Kelley**[11]

[1]Graduate School of Environment Studies, Nagoya University, Nagoya, 464-8601, Japan
[2]Graduate School of Agricultural and Life Sciences, The University of Tokyo, Tokyo, 113-8657, Japan
[3]Japan Agency for Marine-Earth Science and Technology (JAMSTEC), Yokohama, 237-0061, Japan
[4]National Institute for Environmental Studies (NIES), Tsukuba, Ibaraki, 305-8506, Japan
[5]NSF National Center for Atmospheric Research, Boulder, CO, USA
[6]NOAA Geophysical Fluid Dynamics Laboratory, Princeton, NJ, USA
[7]Center for Climate Systems Research, Columbia University, 2880 Broadway, New York, 10025 NY, USA
[8]NASA Goddard Institute for Space Studies, 2880 Broadway, New York, 10025 NY, USA
[9]Norwegian Meteorological Institute, Oslo, Norway
[10]UK Met Office Hadley Centre, Exeter, UK
[11]UK Centre for Ecology & Hydrology, Wallingford, Oxfordshire, OX10 8BB, UK

**Correspondence:** Ngoc Thi Nhu Do (ngocdtn19@gmail.com)

**Abstract.** CE1 Terrestrial isoprene, a biogenic volatile organic compound emitted by many plants, indirectly influences Earth's radiative balance through its interactions with atmospheric oxidants, affecting ozone formation, methane lifetime, and secondary aerosol production. Elucidating its historical changes is therefore important for predicting climate change and air quality. Isoprene emissions can respond to climate (e.g. temperature, shortwave radiation, precipitation), land use and land cover change (LULCC), and atmospheric $CO_2$ concentrations. However, historical trends of isoprene emissions and the relative influences of the respective drivers of those trends remain highly uncertain. This study addresses uncertainty in historical isoprene emission trends and their influential factors, particularly the roles of climate, LULCC, and atmospheric $CO_2$ (via fertilization and inhibition effects). The findings are expected to reconcile discrepancies among different modelling approaches and to improve predictions of isoprene emissions and their climate change effects.

To investigate isoprene emission trends, controlling factors, and discrepancies among models, we analysed long-term (1850–2014) global isoprene emissions from online simulations of CMIP6 Earth system models and offline simulations using the Vegetation Integrative SImulator for Trace gases (VISIT) dynamic vegetation model driven by climate reanalysis data.

Mean annual global present-day isoprene emissions agree well among models (434–510 $TgC\,yr^{-1}$) with a 5 % inter-model spread (24 $TgC\,yr^{-1}$), but regional emissions differ greatly (9 %–212 % spread). All models show an increasing trend in global isoprene emissions in recent decades (1980–2014), but their magnitudes vary ($+1.27 \pm 0.49\ TgC\,yr^{-2}$ CE2, $0.28 \pm 0.11\ \%\,yr^{-1}$). Long-term trends of 1850–2014 show high uncertainty among models ($-0.92$ to $+0.31\ TgC\,yr^{-2}$).

Results of emulated sensitivity experiments indicate meteorological variations as the main factor of year-to-year fluctuations, but the main drivers of long-term isoprene emission trends differ among models. Models without $CO_2$ effects implicate climate change as the driver, but other models with $CO_2$ effects (fertilization only/fertilization and inhibition) indicate $CO_2$ and LULCC as the primary drivers. The discrepancies arise from how models account for $CO_2$ and LULCC alongside climate effects on isoprene emissions. Aside from

LULCC-induced reductions, differences in $CO_2$ inhibition representation (strength and presence or absence of thresholds) were able to mitigate or reverse increasing trends because of rising temperatures or in combination with $CO_2$ fertilization. Net $CO_2$ effects on global isoprene emissions show the highest inter-model variation ($\sigma = 0.43\,\mathrm{TgC\,yr^{-2}}$), followed by LULCC effects ($\sigma = 0.17\,\mathrm{TgC\,yr^{-2}}$), with climate change effects exhibiting more or less variation ($\sigma = 0.06\,\mathrm{TgC\,yr^{-2}}$).

The critical drivers of isoprene emission trends depend on a model's emission scheme complexity. This dependence emphasizes the need for models with accurate representation of $CO_2$ and LULCC effects alongside climate change influences for robust long-term predictions. Important uncertainties remain in understanding the interplay between $CO_2$, LULCC, and climate effects on isoprene emissions, mainly for $CO_2$. More long-term observations of isoprene emissions across various biomes are necessary, along with improved models with varied $CO_2$ responses. Moreover, instead of reliance on the current models, additional emission schemes can better capture isoprene emissions complexities and their effects on climate.

## 1 Introduction

Isoprene (2-methyl-1,3-butadiene, $C_5H_8$), a biogenic volatile organic compound (BVOC) emitted by terrestrial vegetation, strongly shapes our planet's atmospheric chemistry and climate (Fiore et al., 2012). This molecule, accounting for roughly half of global BVOC emissions (Guenther et al., 2012), interacts with atmospheric oxidants, influencing processes such as ozone formation (Arneth et al., 2010; Squire et al., 2014; Wiedinmyer et al., 2006), methane lifetime (Kaplan et al., 2006; Achakulwisut et al., 2015; Hopcroft et al., 2017), and aerosol production (Claeys et al., 2004; Henze and Seinfeld, 2006; Lin et al., 2016; Thornhill et al., 2021; Tsigaridis and Kanakidou, 2018). These processes in turn affect the atmospheric radiative balance. For instance, historical anthropogenic land use and land cover change (LULCC) that decreased BVOC emissions can be expected to have reduced the global formation of secondary organic aerosols (SOAs) by 13 % (Scott et al., 2017), and they have reduced the SOA tropospheric burden by 13 % (Heald and Geddes, 2016) while causing positive radiative forcing (warming effect) of $0.017$–$0.09\,\mathrm{W\,m^{-2}}$ during 1850–2000 through the direct aerosol effect (Heald and Geddes, 2016; Scott et al., 2017; Unger, 2014), with additional positive forcing of $0.008\,\mathrm{W\,m^{-2}}$ from the indirect aerosol effect (Scott et al., 2017). Moreover, isoprene emissions influence the oxidizing capacity of the troposphere by affecting the abundance of the hydroxyl radical (OH) (Karl et al., 2007, 2013). They contribute to cloud formation and precipitation patterns (Boy et al., 2019; Fang et al., 2015; Steiner, 2020). Biogenic isoprene emissions depend strongly on climate (e.g. temperature, shortwave radiation, precipitation), land cover, and atmospheric chemistry (e.g. ambient ozone and $CO_2$ concentrations) (Pacifico et al., 2012), leading to climate feedback (Szopa et al., 2021; Thornhill et al., 2021). Rising temperatures are likely to increase future global BVOC emissions by 30 %–45 % (Peñuelas and Llusià, 2003), potentially leading to a cooling effect through aerosol formation ($-0.06$ to $-0.01\,\mathrm{W\,m^{-2}\,K^{-1}}$) (Paasonen et al., 2013; Scott et al., 2018). Nevertheless, the exact response of isoprene emissions to future changes in climate and $CO_2$ levels remains uncertain (Szopa et al., 2021). Therefore, accurately modelling isoprene emissions and elucidating their response to climate change are crucially important for predicting their roles in air quality and climate.

Sophisticated emission parameterizations including empirical approaches known as Guenther schemes (Guenther, 1997; Guenther et al., 1995), also designated MEGAN (Guenther et al., 2006, 2012) and photosynthesis-based approaches such as the Interactive BVOC (iBVOC) emission scheme (Pacifico et al., 2011), are used to estimate isoprene emissions either offline, using external ground-based or satellite data, or online within regional and global climate–chemistry models. These schemes, which have been developed based on laboratory and field measurements, calculate emissions in each grid cell by incorporating environmental factors (e.g. temperature and photosynthetically active radiation (PAR) – a sub-range of shortwave radiation, precipitation, and atmospheric $CO_2$ concentration), alongside the vegetation distribution and plant-specific emission factors. Regarding $CO_2$ effects, many studies show that higher $CO_2$ concentrations can inhibit isoprene emissions directly, contrary to the expectation of increased emissions from rising temperatures and $CO_2$ fertilization acting on plant growth (Morfopoulos et al., 2014; Naik et al., 2004; Possell et al., 2005; Possell and Hewitt, 2011; Young et al., 2009). However, the responses of isoprene emissions to $CO_2$ concentration vary across plant species (Lantz et al., 2019; Niinemets et al., 2021). Additionally, the $CO_2$ conditions to which the plants were exposed during their growth or acclimation can influence the responses (Possell and Hewitt, 2011; Sun et al., 2013; Wilkinson et al., 2009). Furthermore, leaf temperature plays an important role, with higher temperatures generally dampening the sensitivity of isoprene emissions to elevated $CO_2$ (Monson et al., 2016; Potosnak et al., 2014a; Sun et al., 2013). Future warming, which is expected to include increasing $CO_2$ and temperatures, can primarily affect isoprene emissions. Rising temperatures are expected to boost emission rates, but increasing $CO_2$ concentrations might lower them. These effects were likely the reverse under preindustrial conditions, where lower $CO_2$ concentrations potentially favoured higher emissions (Pacifico et al., 2012; Possell and Hewitt, 2011), although lower temperatures would have led to decreased isoprene emissions (Monson et al., 1992). Additionally, the vegetation distribution, in-

fluenced by both anthropogenic LULCC and climate change, might further reshape global isoprene emissions (Pacifico et al., 2012). During the twentieth century, human activities, particularly changes in land use, played a larger role than natural dynamics in affecting vegetation (Hurtt et al., 2006; Unger, 2013). In fact, land cover change has altered one-third to one-half of Earth's land surface, with large areas of forests converted to cropland (Hurtt et al., 2006; Ma et al., 2020; Vitousek et al., 1997). Despite effectively capturing the short-term response of isoprene emissions to typical environmental fluctuations (Müller et al., 2008; Pacifico et al., 2011; Sindelarova et al., 2014; Weber et al., 2023) and isoprene emission changes under extreme weather events such as drought (characterized by low soil moisture and often accompanied by high temperature and low precipitation) (Jiang et al., 2018; Klovenski et al., 2022; Wang et al., 2022), models are challenged when representing longer-term trends accurately. This difficulty raises important questions about the models' adequacy for predicting the long-term responses of isoprene emissions to $CO_2$, LULCC, and climate variation. Furthermore, the scarcity of long-term direct flux measurements makes comprehensive model validation difficult. Under future climate scenarios with potentially doubled $CO_2$ concentrations and with heavy reliance on mitigation efforts particularly addressing land use change, it is extremely important to examine historical trends in isoprene emissions and to elucidate uncertainties in current models before making predictions about future isoprene emissions.

Few earlier studies have specifically examined historical trends of isoprene emissions and the primary factors driving them. Photosynthesis-based models suggest an increase of 12 %–22 % in emissions because of climate change alone or in combination with $CO_2$ fertilization during the preindustrial era (1901) to recent times (2000) (Arneth et al., 2007a; Hantson et al., 2017; Pacifico et al., 2012; Unger, 2013). However, including the $CO_2$ inhibition effect alone (Arneth et al., 2007a) and in combination with LULCC reverses this trend, leading to a decrease and indicating higher isoprene emissions in the preindustrial era than in recent times (Hantson et al., 2017; Pacifico et al., 2012; Unger, 2013). These earlier reports of the relevant literature describe marked decreases in isoprene emissions from the preindustrial era (1901) to recent times (2000), ranging from −9 % (Arneth et al., 2007a) to −20 % or more (Hantson et al., 2017; Pacifico et al., 2012; Unger, 2013). This trend reversal underscores the importance of including $CO_2$ inhibition (Arneth et al., 2007a; Pacifico et al., 2012) along with LULCC effects (Hantson et al., 2017; Unger, 2013). In contrast, empirical models present a picture that is less clear. A report of an earlier study based on a comparison between $CO_2$ inhibition and temperature factors (Heald et al., 2009) described how $CO_2$ inhibition merely offsets the rising temperature effect on future isoprene emissions but the enhancement of isoprene emissions caused by low ambient $CO_2$ concentrations does not compensate for the effects of cooler tem-

peratures, implying no trend in isoprene emissions over the last 400 000 TS3 years. By contrast, a nearly contemporaneous study (Lathière et al., 2010) using a different $CO_2$ inhibition equation showed a decrease of −16 % with $CO_2$ inhibition, becoming even steeper (−24 %) when LULCC was incorporated. This finding contrasts with the +7 % increase estimated for 1901–2000, when only climate and $CO_2$ fertilization were included. Results reported by Lathière et al. (2010) align with some earlier studies conducted using photosynthesis-based models. However, Tanaka et al. (2012) reported that isoprene emissions were reduced by only −2 % during 1850–2000 because of radiation, suggesting that rising temperature effects compensated for the LULCC effects when both $CO_2$ effects were not considered. Therefore, the larger picture of changes in isoprene emission remains unclear, with reports describing decreases ranging from a modest −2 % (Tanaka et al., 2012) to a remarkable −20 % or more (Arneth et al., 2007a; Hantson et al., 2017; Lathière et al., 2010; Pacifico et al., 2012; Unger, 2013) or even increases of 7 %–12 % (Arneth et al., 2007a; Lathière et al., 2010). These discrepancies likely arise from differences in research methods, including emission model approaches (photosynthesis-based vs. empirical), climate models, and the representation of $CO_2$ effects (fertilization vs. inhibition) and LULCC. Additionally, during historic periods, the dominant drivers of changes in isoprene emission remain unclear. Whereas Heald et al. (2009) implied temperature was the primary controlling factor throughout the historical period, Unger (2013) and Hantson et al. (2017) argued that LULCC stands as the primary driver. Other studies (Lathière et al., 2010; Pacifico et al., 2012) have proposed that $CO_2$ inhibition, in addition to LULCC, can play an important role in isoprene emission changes. Although $CO_2$, LULCC, and climate are recognized as key drivers, their effects on long-term isoprene emission trends demand further investigation.

From just a single Earth system model (GISS-ModelE2) (Tsigaridis and Kanakidou, 2018) in the Coupled Model Intercomparison Project Phase 5 (CMIP5), the latest CMIP6 now includes multiple coupled climate–chemistry models and Earth system models (ESMs) that provide online simulations of isoprene emissions, thereby enabling the first comparison of simulations within a consistent framework. Building upon an earlier study specifically addressing future simulations (Cao et al., 2021), the present study examines historical trends, comprehensively analysing isoprene emission patterns during 1850–2014, which corresponds to the CMIP6 historical period. The goals of this study are the following: (1) investigate long-term global isoprene emission trends; (2) identify the dominant drivers of these respective trends, including $CO_2$, LULCC, and physical climate factors; and (3) analyse the causes of uncertainties in current model simulations. We applied random forest regression to emulate sensitivity experiments of isoprene emissions and ascertain the critical drivers of isoprene emission trends in the online CMIP6 models. We subsequently compared the results of

this analysis to those obtained from sensitivity experiments using the Vegetation Integrative SImulator for Trace gases (VISIT) offline land-surface model (Inatomi et al., 2010; Ito, 2019a). Our inclusion of VISIT provides two key benefits: computational efficiency and model assessment. Running numerous sensitivity experiments within complex CMIP6 models can be resource-intensive, but VISIT, a simpler offline model emphasizing land-surface processes, facilitates efficient exploration of isoprene emissions sensitivity to various factors. By comparing the key drivers identified in CMIP6 models with the results obtained from a well-understood model such as VISIT, we can gain a more comprehensive understanding and can support findings obtained from the complex CMIP6 models. By addressing these fundamental questions, this study can elucidate the role of isoprene in a changing climate and can contribute to the development of more accurate and reliable ESMs. The following section presents a description of the VISIT simulations, CMIP6 datasets, and statistical methods used for this study. A comparison of isoprene emission trends and their attributions is explored in Sect. 3. Then uncertainties in isoprene emission trends and suggestions for future development are discussed in Sect. 4. Finally, the main conclusions inferred from the results are summarized in Sect. 5.

## 2 Data and methods

### 2.1 VISIT data

#### 2.1.1 VISIT model overview

VISIT is a process-based terrestrial ecosystem model simulating carbon, nitrogen, and water cycles (Inatomi et al., 2010; Ito, 2010). Its hydrology submodule uses forcing meteorological data (incoming radiation, precipitation, temperature, humidity, vapour pressure, and cloudiness) and biophysical properties (vegetation cover, albedo, and soil water-storage capacity) to simulate land-surface radiation and water budgets (Ito, 2019a). The model comprises plant and soil components in an ecosystem, allowing for the integrated simulation of land–atmosphere biogeochemical interactions.

The carbon cycle within VISIT inherits the foundation laid by the Sim-CYCLE model (Ito and Oikawa, 2002) and encompasses key processes parameterized by $CO_2$ concentration, temperature, radiation, and water. Photosynthesis, which is responsible for most plant gross primary productivity (GPP), is simulated based on Monsi–Saeki theory (Monsi and Saeki, 1953), allowing for scaling leaf-level photosynthesis to estimate canopy-level primary production (Hajima et al., 2020). Leaf phenology for deciduous forests and grasslands is estimated using an empirical procedure based on the threshold cumulative temperature (Ito, 2019a), leading to improved GPP estimation (Ito and Ichii, 2021). The leaf area index (LAI) and mass are then updated in response to phenological stages and net carbon assimilation. The VISIT model has further expanded capabilities because it incorporates the nitrogen cycle (e.g. $N_2O$ emissions from the soil surface) and trace-gas-related processes (e.g. $CH_4$ emissions from wetland and BVOC emissions). The VISIT model has undergone extensive evaluation of its carbon cycle simulations across various scales from point scales (Hirata et al., 2014; Inatomi et al., 2010; Ito and Oikawa, 2002) to regional scales (Ito and Ichii, 2021). Moreover, the model has been examined through model comparison projects (e.g. Tian et al., 2015; Huntzinger et al., 2017). The VISIT model is also coupled with the MIROC-CHASER atmosphere and chemistry model (Ha et al., 2021; He et al., 2022; Sekiya et al., 2018; Sudo et al., 2002) and COCO (CCSR Ocean Component Model; Hasumi, 2006) to build the Earth system model (Hajima et al., 2020), but it can be run alternatively as a stand-alone model.

The VISIT model incorporates the Guenther scheme (Guenther, 1997) (G1997), designated VISIT(G1997), to estimate BVOC emissions, including those of isoprene. This scheme calculates the emission rate $E_i$ ($\mu$gC m$^{-2}$ per month) based on the following equation.

$$E_i = \mathrm{EF}_i \times \mathrm{EA} = \mathrm{EF}_i \times \mathrm{FM}_{(CO_2\_fert)} \times \gamma_{\mathrm{TMP}}$$
$$\times \gamma_{\mathrm{PPFD}} \times \gamma_{\mathrm{A}} \times \gamma_{\mathrm{CE}} \tag{1}$$

In that equation, $\mathrm{EF}_i$ ($\mu$gC g$_{mass}^{-2}$ h$^{-1}$) is the emission factor of isoprene applied for each plant functional type (PFT) in standard temperature conditions (303.15 K). These values were derived from an earlier study by Lathière et al. (2006). The emission activity factor (EA) accounts for variations in emissions attributable to environmental and phenological factors. The foliar mass (FM; g$_{mass}$C) is calculated by multiplying the average foliar density for $C_3$ and $C_4$ plants (g$_{mass}$C m$^{-2}$) by the day length per month (hour) within VISIT(G1997). In addition, $\gamma_{\mathrm{TMP}}$, $\gamma_{\mathrm{PPFD}}$, $\gamma_{\mathrm{A}}$, and $\gamma_{\mathrm{CE}}$ are activity factors representing the dependence of isoprene emissions on temperature, light (photosynthetic photon flux density), leaf age, and the canopy environment, respectively. The effects of leaf age on isoprene emissions differ between evergreen and deciduous vegetation types. To account for this difference, the model incorporates a modified leaf age distribution based on its simulations. This approach assigns values between 0.05 for immature leaves (less than 1 month old) and 1.2 for mature leaves (2–10 months for deciduous and 3–24 months for evergreen plants). Consequently, the model captures emission reduction caused by leaf senescence by decreasing the $\gamma_{\mathrm{A}}$ value (Ito, 2019a). It is noteworthy that the current version of VISIT(G1997) includes the fertilization effect on photosynthesis only: simulated BVOC emissions respond to $CO_2$ indirectly through the change in leaf mass or LAI. It does not account for the direct inhibition effect of $CO_2$ on isoprene emissions. Precipitation also indirectly affects isoprene emissions via its effects on photosynthesis, which subsequently changes LAI.

### 2.1.2 VISIT(G1997) simulations

This study used the stand-alone version of VISIT(G1997) to simulate global isoprene emissions during 1700–2021 at a spatial resolution of $0.5° \times 0.5°$ at monthly intervals. We incorporated historical $CO_2$ concentrations derived from ice cores and NOAA observations prepared for TRENDY and the Global Carbon Project (Friedlingstein et al., 2022), as well as a land cover dataset with 16 plant types (Ramankutty and Foley, 1999). Details of both datasets are described elsewhere (Ito, 2023). Land use change data from LUH2 data (Hurtt et al., 2020) were also used. Meteorological data for 1901–2021 (temperature, precipitation, vapour pressure, and cloudiness) were taken from CRU TS 4.06 (Harris et al., 2020).

To assess the degree to which different factors influence isoprene emission trends, we conducted the seven sensitivity simulations (S0–S6) presented in Table 1. Each simulation involved a spin-up phase lasting 300–3000 years, depending on the biome type for each grid cell, initialized with 1700 $CO_2$, 1901 climate, and constant 1700 land use data and followed by two transient periods:

- 1700–1900 – varied $CO_2$ concentration and fixed climate (as in spin-up) in S1 and S2 and additionally varied LULCC in S3;

- 1901–2021 – varied $CO_2$ concentration and fixed climate in S1, varied $CO_2$ concentration and climate in S2, and additionally varied LULCC in S3.

In S0, all forcing data are held constant with the $CO_2$ concentration and LULCC fixed in 1850 and climate fixed in 1901 for the two periods. Three additional experiments (S4, S5, S6) isolated the effects of individual climate drivers (temperature, radiation, and precipitation) by holding them constant at 1901 levels while allowing other factors to vary. The VISIT(G1997) model outputs for 1850–2014 were then extracted for comparison with those from CMIP6 models.

### 2.2 CMIP6 data

#### 2.2.1 Model description

We analysed data obtained from the *historical* experiment in which all forcings (e.g. greenhouse gases (GHGs), aerosols, land use, solar, volcanic aerosols) evolved during 1850–2014 (Eyring et al., 2016). Relevant to this work, the external forcings for those simulations included land use change data originating from LUH2 (Hurtt et al., 2020), anthropogenic emissions from CEDS (Hoesly et al., 2018), and biomass burning emissions from BB4CMIP (van Marle et al., 2017). Five ESMs from CMIP6 were selected for their online BVOC emission schemes, including isoprene, but other ESMs use prescribed (interannually fixed) BVOC emissions (Gomez et al., 2023). Among the five selected ESMs, four used the empirically based Guenther scheme for isoprene estimation, albeit using different versions: CESM2-WACCM and NorESM2-LM share the same land component (Community Land Model – CLM5), employing the latest version of MEGAN, MEGANv2.1 (G2012) (Guenther et al., 2012), whereas GFDL-ESM4 and GISS-E2.1-G rely on earlier versions (G2006 and G1995, respectively) (Guenther et al., 1995, 2006). It is noteworthy that UKESM1-0-LL adopted the distinct photosynthesis-based iBVOC scheme (P2011) (Pacifico et al., 2011) to estimate isoprene emissions based on temperature, $CO_2$ concentration, and GPP. Actually CE3, iBVOC in UKESM1-0-LL is derived from the Arneth et al. (2007b) and Niinemets et al. (1999 TS4) models, linking photosynthesis-derived electrons to the isoprene production rate, which is light-dependent. Because UKESM1-0-LL does not simulate electron transport directly, GPP is used as an approximation. Temperature and $CO_2$ inhibition effects are included empirically, accounting for differences in temperature optima and $CO_2$ responses between photosynthesis and isoprene synthesis. Table 2 presents simplified equations of isoprene emission schemes. Table S1 in the Supplement presents descriptions of symbols and parameters. Detailed model descriptions of other related processes are available elsewhere in the literature (Bauer et al., 2020; Emmons et al., 2020; Horowitz et al., 2020; Seland et al., 2020; Sellar et al., 2019).

Generally speaking, temperature and shortwave radiation enhance isoprene emissions in all models despite their differing component structures (Cao et al., 2021). Precipitation effects are represented either by the soil moisture factor in MEGAN (e.g. NorESM2-LM(G2012)) or indirectly through photosynthesis (GPP) in the UKESM1-0-LL(P2011) model (Clark et al., 2011). The influence of $CO_2$ on isoprene emissions varies across models. GFDL-ESM4(G2006) and GISS-E2.1-G(G1995) neglect both direct $CO_2$ inhibition and indirect fertilization effects because of prescribed satellite LAI (Horowitz et al., 2020; Ito et al., 2020). Also, CESM2-WACCM(G2012) and NorESM2-LM(G2012) use direct $CO_2$ inhibition effect parameterization from Heald et al. (2009), whereas UKESM1-0-LL(P2011) uses the scheme from Arneth et al. (2007b). The latter three models also incorporate indirect $CO_2$ fertilization effects through vegetation growth and terrestrial carbon processes, influencing LAI or GPP.

#### 2.2.2 Emulation of isoprene emissions

Understanding the complex influences on isoprene emissions requires analyses of various factors. The initial step is to reproduce isoprene emissions for each ESM accurately based on key input factors including atmospheric $CO_2$ concentration, LULCC, and climate variables. To accomplish this step, we developed a data-driven regression model using a machine learning approach to estimate annual isoprene emissions based on critical drivers within

**Table 1.** Summary of VISIT(G1997) simulations for the studied period (1850–2014).

| Simulation no. | CO$_2$ conc. | LULCC | Climate | | |
| --- | --- | --- | --- | --- | --- |
| | | | Temperature | Shortwave radiation | Precipitation |
| S0 | Fixed in 1850 | Fixed in 1850 | Climate fixed in 1901 | | |
| S1 | – | Fixed in 1850 | | | |
| S2 | – | Fixed in 1850 | – | – | – |
| S3 | – | – | – | – | – |
| S4 | – | – | Fixed in 1901 | – | – |
| S5 | – | – | – | Fixed in 1901 | – |
| S6 | – | – | – | – | Fixed in 1901 |

"–" denotes a variable that varied annually during the simulation period.

**Table 2.** Summary of CMIP6 models and their simplified isoprene emission schemes.

| Model (variant) | Institute (country) CE4 | Resolution (lat, long) | Scheme | Simplified equation |
| --- | --- | --- | --- | --- |
| CESM2-WACCM (r1i1p1f1) | NCAR (USA) | 0.9° × 1.25° | G2012 | $E_i = \text{EF}_i \times \text{LAI}_{\text{dynamic(CO}_2\_\text{fert})} \times \gamma_{\text{CO}_2\_\text{inhi}} \times \gamma_{\text{TMP}} \times \gamma_{\text{PPFD}} \times \gamma_{\text{A}} \times \gamma_{\text{CE}}$ |
| NorESM2-LM (r1i1p1f1) | NCC (Norway) | 1.9° × 2.5° | G2012 | $E_i = \text{EF}_i \times \text{LAI}_{\text{dynamic(CO}_2\_\text{fert})} \times \gamma_{\text{CO}_2\_\text{inhi}} \times \gamma_{\text{TMP}} \times \gamma_{\text{PPFD}} \times \gamma_{\text{SM}} \times \gamma_{\text{A}} \times \gamma_{\text{CE}}$ |
| GFDL-ESM4 (r1i1p1f1) | NOAA (USA) | 1° × 1.25° | G2006 | $E_i = \text{EF}_i \times \text{LAI}_{\text{prescribed(noCO}_2\_\text{fert})} \times \gamma_{\text{TMP}} \times \gamma_{\text{PPFD}} \times \gamma_{\text{A}} \times \gamma_{\text{CE}}$ |
| GISS-E2.1-G (r1i1p3f1) | NASA (USA) | 2° × 2.5° | G1995 | $E_i = \text{EF}_i \times \text{LAI}_{\text{prescribed(noCO}_2\_\text{fert})} \times \gamma_{\text{TMP}} \times \gamma_{\text{PPFD}}$ |
| UKESM1-0-LL (r1i1p1f2) | MOHC (UK) | 1.25° × 1.88° | P2011 | $E_i = \text{EF}_i \times \text{GPP}_{(\text{CO}_2\_\text{fert,PPFD,SM})}/\text{GPP}_{\text{st}} \times \gamma_{\text{CO}_2\_\text{inhi}} \times \gamma_{\text{TMP}}$ |

$E_i$ is the isoprene emission rate. $\text{EF}_i$ is the isoprene emission factor applied for each PFT under standard conditions. $\text{LAI}_{\text{dynamic(CO}_2\_\text{fert})}$ is the leaf area index updated in response to increasing $CO_2$ concentration via photosynthesis ($CO_2$ fertilization effect). $\text{LAI}_{\text{prescribed(noCO}_2\_\text{fert})}$ is the prescribed satellite leaf area index which does not consider $CO_2$ fertilization effect. GPP denotes gross primary productivity, updated in response to increasing $CO_2$ concentration via photosynthesis ($CO_2$ fertilization effect), light, and soil moisture. "st" represents standard conditions. $\gamma_{\text{CO}_2\_\text{inhi}}$, $\gamma_{\text{TMP}}$, $\gamma_{\text{PPFD}}$, $\gamma_{\text{SM}}$, $\gamma_{\text{A}}$, and $\gamma_{\text{CE}}$ stand for activity factors representing the $CO_2$ inhibition effect and dependence of isoprene emissions caused by temperature, light (photosynthetic photon flux density), soil moisture, leaf age, and the canopy environment, respectively. Detailed descriptions of symbols and parameters in the isoprene emission equations are presented in Table S1.

ESMs. For each ESM, the features input into the regression model include the same drivers for isoprene emission simulation used in the CMIP6 project. For models that are sensitive to $CO_2$ (CESM2-WACCM(G2012), NorESM2-LM(G2012), UKESM1-0-LL(P2011)), the features comprise $CO_2$; LULCC (represented by the tree fraction); and climate variables such as temperature (tas), surface downwelling shortwave radiation (rsds; hereafter referred to as shortwave radiation), and precipitation (pr). For those models which are not sensitive to $CO_2$ (GFDL-ESM4(G2006) and GISS-E2.1-G(G1995)), the features only include LULCC and climate variables. We chose random forest as the machine learning algorithm to build regression models because of its ability to handle complex, nonlinear relations among variables without requiring assumptions (Guo et al., 2015; Zhang et al., 2017).

To assess the regression model performance, we used a threefold cross-validation approach. The dataset was divided into three periods: 1850–1904 (fold 1), 1905–1959 (fold 2),

and 1960–2014 (fold 3). During each iteration, one fold was designated as the test set, whereas the remaining two folds were used for training the model. The evaluation metrics included the coefficient of determination ($R^2$), root mean square error (RMSE), and mean absolute error (MAE). The results of the threefold cross-validation for each ESM are presented in Fig. S1 in the Supplement, in which the models exhibited high $R^2$ values ($> 0.9$) and low errors across all ESMs during cross-validation. Furthermore, the estimated historical trend of global annual isoprene emission using random forest showed strong correlation ($r > 0.9$) and consistency with the CMIP6 simulation for all ESMs (Fig. S2). These results demonstrate that the random forest models perform well in reproducing isoprene emissions for all CMIP6 models based on the selected variables.

For further identification of the individual effects of each driver on isoprene emissions, we used random forest regression as an emulator to replicate sensitivity experiments for

each CMIP6 model. This method involved generation of isoprene emissions using the trained random forest regressor with varied settings for the input drivers for each CMIP6 ESM, as outlined in Tables S2 and S3. First, to investigate the effects of $CO_2$, LULCC, and climate on isoprene emissions, we emulated sensitivity experiments (S0–S3) similar to those conducted using the VISIT(G1997) model for $CO_2$-sensitive models including CESM2-WACCM(G2012), NorESM2-LM(G2012), and UKESM1-0-LL(P2011). For the other two models, GFDL-ESM4(G2006) and GISS-E2.1-G(G1995), we modified the experiments (S1′–S3′) to compare the LULCC and climate effects. Subsequently, to identify the main climate drivers (temperature, shortwave radiation, and precipitation) of isoprene emissions, we conducted three additional simulations similar to S4–S6 of VISIT(G1997). Detailed descriptions of these experiments are presented in Tables S2 and S3.

All historical simulation data (including isoprene emissions and other relevant variables described above) for each model were downloaded via the Earth System Grid Federation portal (ESGF, 2023). To facilitate comparison, we resampled spatial resolutions of different models to a consistent $1° \times 1.25°$ grid and calculated annual values before analysis.

## 2.3 Analysis methods

### 2.3.1 Attributions of isoprene emission changes

To isolate the effects of $CO_2$, LULCC, and climate on isoprene emissions, four simulations (S0, S1, S2, and S3) were conducted for VISIT(G1997) and CMIP6 models to consider the $CO_2$ effects (CESM2-WACCM(G2012), NorESM2-LM(G2012), and UKESM1-0-LL(P2011)). Although VISIT only incorporates the $CO_2$ fertilization effect (increase) via LAI/GPP CE5, these CMIP6 models also account for the direct $CO_2$ inhibition effect (decrease) on isoprene emissions. Consequently, in VISIT, the $CO_2$ effects on isoprene emissions are attributable to fertilization only. By contrast, in these CMIP6 models, the net effect of both fertilization and inhibition effects is included. The effects were calculated by comparing isoprene emission simulations with fixed and varied states for each driver: the $CO_2$ effect (difference in isoprene emissions between S1 and S0, S1 − S0 CE6), the LULCC effect (S3 − S2), the climate effect (S2 − S1), and three combined drivers (S3 − S0). For models without the $CO_2$ effects, GFDL-ESM4(G2006) and GISS-E2.1-G(G1995), three simulations (S1′, S2′, and S3′) were used, with effects calculated as the LULCC effect (S3′ − S2′), climate effect (S2′ − S1′), and two combined drivers (S3′ − S1′).

Finally, to identify individual climate driver effects (temperature, shortwave radiation, and precipitation), we compared S3 (all varied) with three additional fixed-case simulations (S4, S5, and S6): the temperature effect (S3 − S4), the radiation effect (S3 − S5), and the precipitation effect (S3 − S6).

### 2.3.2 Significance test for long-term trends of isoprene emissions

We performed the Mann–Kendall trends test (Kendall, 1975; Mann, 1945) and the Theil–Sen estimator (Sen, 1968; Theil, 1950) to detect robust trends in annual isoprene emissions and their attributing factors (i.e. the effects of $CO_2$, LULCC, and climate (temperature, radiation, and precipitation)) for each model at global and grid scales. The Mann–Kendall test, a nonparametric method for trend analysis, has been employed widely for analysing hydrometeorological and biogeochemical time series (Kondo et al., 2018; Pan et al., 2020; Yue and Wang, 2004). The Theil–Sen method was used to calculate the magnitude of the trend. The Mann–Kendall test was used to determine the significance level ($p < 0.05$) of the trends in isoprene emissions.

At a global scale, trend tests were conducted for 1850–2014, 1850–1979, and 1980–2014, applied to global annual totals of isoprene emission and global means of climate variables from historical simulation of CMIP6 models and VISIT-S3(G1997) over land area (Table 3). At the grid scale, similar trend tests were applied for 1850–2014 to annual totals of isoprene emissions (Fig. 5) and to annual means of climate variables from historical simulation of CMIP6 models and VISIT-S3(G1997) (Fig. S7).

To identify trends in contributing factors to global isoprene emissions (1850–2014), we used the effects of $CO_2$, LULCC, climate, and individual climate drivers (as presented in Sect. 2.3.1) from CMIP6 and VISIT(G1997) model simulations. Trend tests were then applied to individual model outputs (Figs. 7 and 9). For grid-scale trends (1850–2014), we applied trend tests to annual isoprene emissions from the effects of each driver ($CO_2$, LULCC, and climate) (Fig. 10). The driver with the absolute most prominent isoprene emission trend was identified as the dominant driver (Fig. 11). Similarly, to identify the main climate driver for grid-scale trends (1850–2014), we applied the trend test to annual isoprene emissions induced by each climate driver (Fig. 12). The climate factor with the absolute most prominent isoprene emission trend was then identified as the dominant climate driver (Fig. 13).

## 3 Results

### 3.1 Global and regional isoprene emissions in the present day

The mean annual global isoprene emissions during 2000–2014 agreed well with results obtained using VISIT-S3(G1997) simulation and CMIP6 models, at 434–510 TgC yr$^{-1}$ (Fig. 1a). GFDL-ESM4(G2006) stands at the lower end with 434 TgC yr$^{-1}$, whereas VISIT-S3(G1997) of-

fers the highest estimate at $510\,\mathrm{TgC\,yr^{-1}}$. These values fall within the broader range of $308\text{–}705\,\mathrm{TgC\,yr^{-1}}$ (equivalent to $350\text{–}800\,\mathrm{Tg\,yr^{-1}}$) described in reports of earlier studies using stand-alone MEGAN models with diverse weather and land cover data for 2000 (Guenther et al., 2012). Recent studies using an empirical approach with MEGANv2.1(G2012) further support this range, estimating the mean annual global isoprene emissions for the present day as $370\text{–}524\,\mathrm{TgC\,yr^{-1}}$. Examples include $370\,\mathrm{TgC\,yr^{-1}}$ for MEGAN-MOHYCAN covering 2000–2016 (Opacka et al., 2021), $388\,\mathrm{TgC\,yr^{-1}}$ for CAMS-GLOB-BIOv3.1 over 2000–2019 (Sindelarova et al., 2022), and $524\,\mathrm{TgC\,yr^{-1}}$ for MEGAN-MACC during 1980–2010 (Sindelarova et al., 2014). It is noteworthy that all these datasets except MEGAN-MACC neglected the effects of $CO_2$ inhibition and soil moisture, which MEGAN-MACC does account for (Sindelarova et al., 2014). Earlier studies using a photosynthesis-based approach also reported diverse estimates (Pacifico et al., 2011) CE7, reporting global annual emissions of $516\text{–}522\,\mathrm{TgC\,yr^{-1}}$ for 1990–1999 and $463\,\mathrm{TgC\,yr^{-1}}$ for 1981–2002 (Arneth et al., 2011). In this study's present day (2000–2014), the UKESM1-0-LL(P2011) isoprene emission of $478\,\mathrm{TgC\,yr^{-1}}$ aligns with the range of these earlier studies.

Although CMIP6 models and VISIT-S3(G1997) simulation agreed well in terms of the mean annual global isoprene emissions over 2000–2014, pronounced differences were found for regional emissions, as presented in Fig. 1b and in Table S4. The tropics dominate isoprene emissions across all models, but the contribution varies considerably. The mean annual emissions for tropical regions including the Amazon (AMZ), West Africa (WAF), East Africa (EAF), and Southeast Asia (SEA), are $207\text{–}377\,\mathrm{TgC\,yr^{-1}}$, representing 43 % to 73 % of global isoprene emissions for GISS-E2.1-G(G1995) and CESM2-WACCM(G2012)/NorESM2-LM(G2012). Other models estimate a similar tropical dominance, exceeding 50 % of global totals.

Considerable uncertainties were found for tropical isoprene emissions, particularly in the Amazon. Model-derived estimates vary from $98\,\mathrm{TgC\,yr^{-1}}$ in GISS-E2.1-G(G1995) to $175\,\mathrm{TgC\,yr^{-1}}$ in CESM2-WACCM(G2012) and NorESM2-LM(G2012). Similarly, Southeast Asia displays wide variation, with GISS-E2.1-G(G1995) and NorESM2-LM(G2012) estimated as $14.7\text{–}87.8\,\mathrm{TgC\,yr^{-1}}$. However, West Africa (WAF) shows less variation, with model estimates ranging from $45.5\text{–}68.6\,\mathrm{TgC\,yr^{-1}}$ and GFDL-ESM4(G2006) offering the lowest estimate.

Arid and semiarid regions display the highest variability in isoprene emission estimates. For instance, GISS-E2.1-G(G1995) attributes considerably higher emissions to the Sahara (SAH) ($24.9\,\mathrm{TgC\,yr^{-1}}$) than other models do ($0.1\text{–}4.3\,\mathrm{TgC\,yr^{-1}}$). Similarly, in northern Australia (NAU), GISS-E2.1-G(G1995) estimates emissions of $49.8\,\mathrm{TgC\,yr^{-1}}$, which is approximately 2–7 times higher than those estimates of other models ($6.4\text{–}27.5\,\mathrm{TgC\,yr^{-1}}$). Furthermore, isoprene emissions remain consistent across models in east-

ern North America (ENA) ($5.1\text{–}6.7\,\mathrm{TgC\,yr^{-1}}$) and East Asia (EAS) ($15.2\text{–}21.6\,\mathrm{TgC\,yr^{-1}}$). For northeastern Brazil (NEB), VISIT-S3(G1997) calculates an emission of $30.9\,\mathrm{TgC\,yr^{-1}}$, which is closely aligned with UKESM1-0-LL(P2011)'s estimate of $27.67\,\mathrm{TgC\,yr^{-1}}$ but exceeds those estimates in other models ($8.7\text{–}13.2\,\mathrm{TgC\,yr^{-1}}$).

Across all models, isoprene emissions exhibit an apparent decline from warm and humid tropical forests towards colder and drier biomes such as tundra and deserts, as presented in Fig. 2. However, large discrepancies are apparent between model estimates in tropical regions (the Amazon, equatorial Africa, and Southeast Asia). The Amazon consistently stands out as the region with the highest isoprene emissions across all models, but the magnitude of this emission varies considerably. The CESM2-WACCM(G2012) and NorESM2-LM(G2012) models simulate intense emissions, with the central Amazon showing the highest values. VISIT-S3(G1997) and UKESM1-0-LL(P2011) also identify the central Amazon as an emission hotspot but show relatively high emissions in the northwestern Amazon. This pattern aligns with satellite retrievals of isoprene concentrations over this region (Wells et al., 2022), though the emissions are of a smaller magnitude compared to those simulated by CESM2-WACCM(G2012) and NorESM2-LM(G2012). However, the GFDL-ESM4(G2006) and GISS-E2.1-G(G1995) models recognize the western Amazon and northern Amazon as emission hotspots, respectively, although these signals are weaker compared to in other models. Equatorial Africa, including West Africa (WAF) and East Africa (EAF), presents a contrasting scenario. Actually, GISS-E2.1-G(G1995) suggests a broader area of high emission, but other models concentrate this peak around $20°\,\mathrm{E}$. The spatial emission pattern in Southeast Asia remains consistent across models, but CESM2-WACCM(G2012) and NorESM2-LM(G2012) show higher values. Low emission estimates consistently characterize high northern latitudes (e.g. Alaska (ALA); Canada, Greenland, and Iceland (CGI)) and arid–semiarid regions such as Central Asia (CAS), western North America (WNA), and the Sahara (SAH). Northern Australia (NAU) is a notable exception, with GISS-E2.1-G(G1995) estimating much higher emissions than other models, which offer consistently low values.

As presented in Fig. 3, the latitudinal profiles of isoprene emissions reveal general agreement among models, with a single peak around the Equator. The exception is GISS-E2.1-G(G1995), which exhibits a second peak around $-25°\,\mathrm{S}$, coinciding with high emissions in Australia. Two models, CESM2-WACCM(G2012) and NorESM2-LM(G2012) stand out for their steeper spatial gradient, particularly between the tropics and other regions, showing lower emissions at high latitudes and higher emissions in the tropics than other models do. A considerable degree of uncertainty prevails within the tropics, with the highest and lowest estimates differing nearly 2-fold. CESM2-WACCM(G2012) and NorESM2-LM(G2012) estimate the highest emissions, ex-

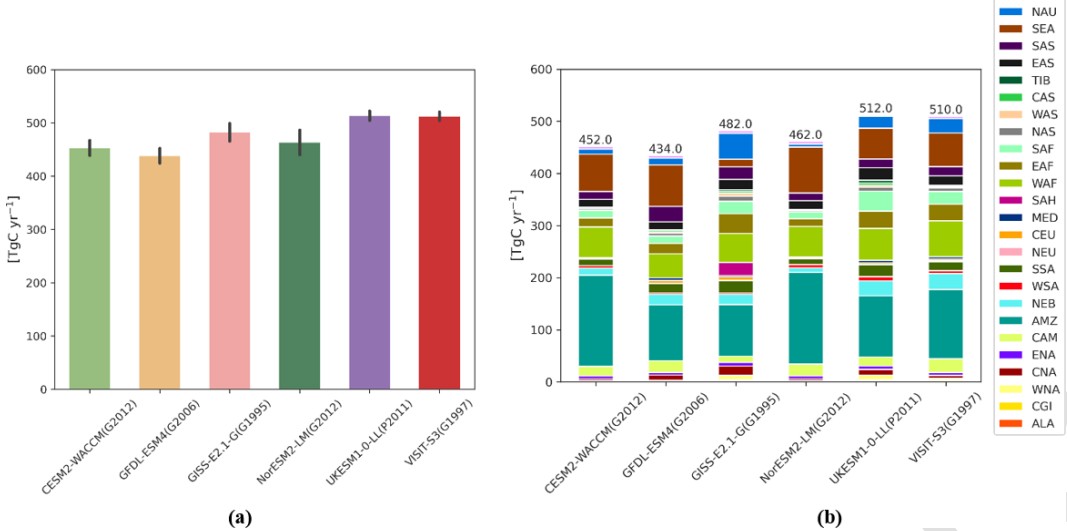

**Figure 1.** Mean annual **(a)** global and **(b)** regional isoprene emissions in the present day (2000–2014). Error bars in **(a)** represent the standard deviation for each model. The top of **(b)** portrays the absolute contributions of 26 regions to global totals. These regions correspond to the 26 SREX regions defined by the IPCC Special Report called *Managing the Risks of Extreme Events and Disasters to Advance Climate Change Adaptation* (Seneviratne et al., 2012), as shown in Fig. S3. The colour bar on the right shows the colours assigned to the respective regions.

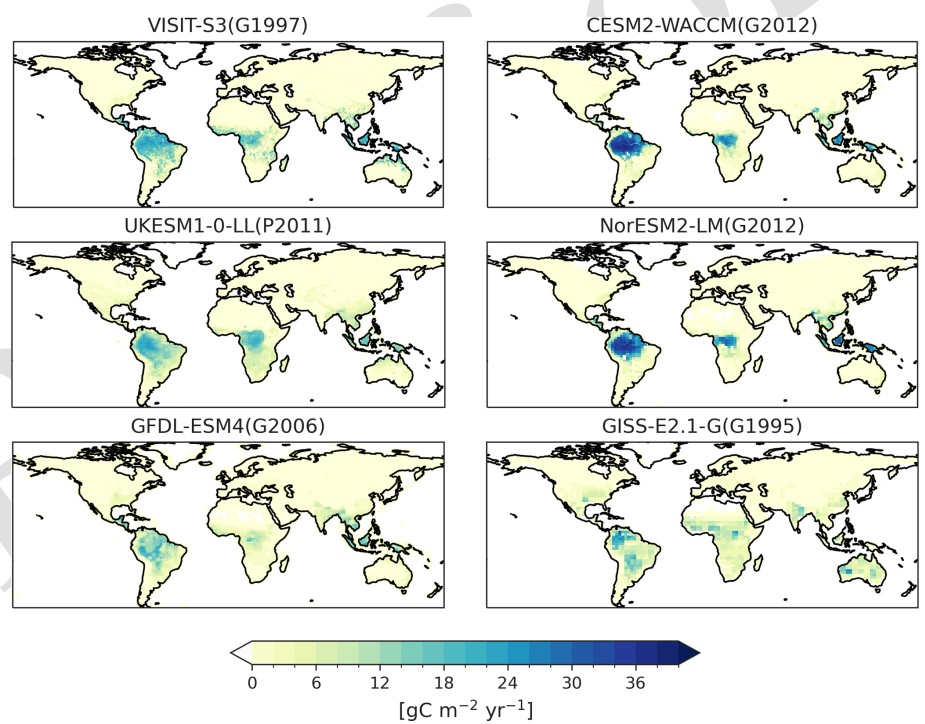

**Figure 2.** Spatial distributions of mean annual isoprene emissions in the present day (2000–2014).

ceeding $4\,\mathrm{gC\,m^{-2}\,yr^{-1}}$, whereas GISS-E2.1-G(G1995) estimates much lower values below $2.5\,\mathrm{gC\,m^{-2}\,yr^{-1}}$. VISIT-S3(G1997) and UKESM1-0-LL(P2011) occupy the middle range, with estimates of approximately $3\,\mathrm{gC\,m^{-2}\,yr^{-1}}$. The primary reason underlying variation in the regional distribution of isoprene emissions among models likely stems from discrepancies in vegetation distribution and emission factors, as elaborated in Sect. 4.1.1.

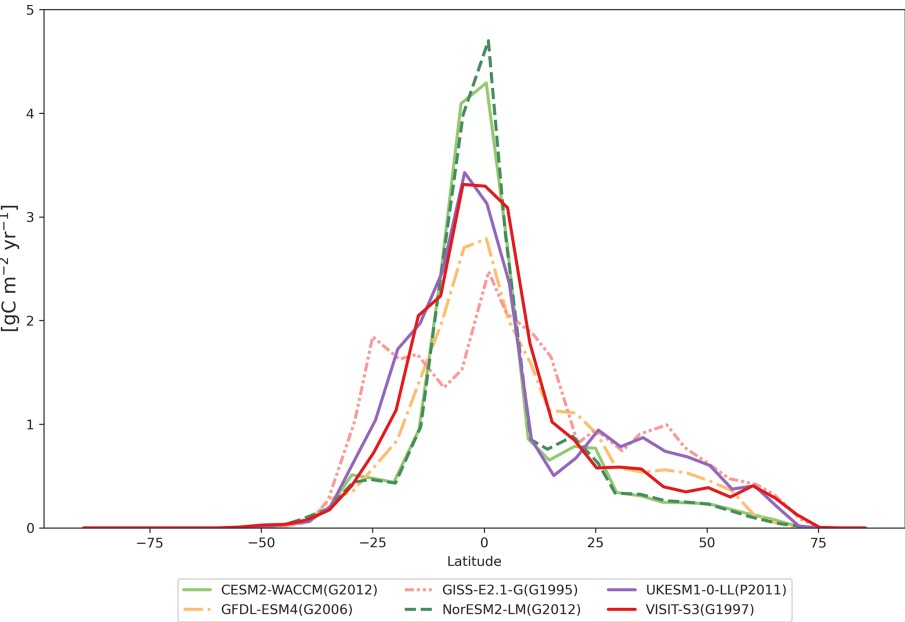

**Figure 3.** Latitudinal profiles of mean annual isoprene emissions in the present day (2000–2014).

## 3.2 Long-term trends of global isoprene emissions

Considerable uncertainty surrounds global isoprene emission trends throughout 1850–2014 (Table 3 and Fig. 4). While three models (VISIT-S3(G1997), GFDL-ESM4(G2006), and GISS-E2.1-G(G1995)) show an upward trend ($+0.13$ to $+0.31$ TgC yr$^{-2}$), CESM2-WACCM(G2012) and NorESM2-LM(G2012) find no significant trend. In contrast, UKESM1-0-LL(P2011) exhibits a decline in global isoprene emissions at a rate of $-0.92$ TgC yr$^{-2}$. Despite these differences in isoprene emission trends, all models show that temperature increases consistently: $+0.002$ to $+0.005\,°C$ yr$^{-1}$.

From preindustrial times to 1980 includes a period of considerable uncertainty in isoprene emissions, despite most models using similar algorithms, except UKESM1-0-LL(P2011) (Table 2). Three models (CESM2-WACCM(G2012), NorESM2-LM(G2012), and UKESM1-0-LL(P2011)) show a significant decrease ($p < 0.05$), with UKESM1-0-LL(P2011) experiencing the largest decline of $-0.92$ TgC yr$^{-2}$ compared to a $-0.11$ TgC yr$^{-2}$ decrease in the other two models. Conversely, GFDL-ESM4(G2006), GISS-E2.1-G(G1995), and VISIT-S3(G1997) show significant increases ($+0.07$ to $+0.13$ TgC yr$^{-2}$). For temperature, except for NorESM2-LM(G2012) and UKESM1-0-LL(P2011), which revealed no significant trend, the other models exhibited a significant and moderate trend of $+0.002\,°C$ yr$^{-1}$ for this period (1850–1979).

From 1980–2014, all models show significantly increasing trends in global isoprene emissions, but their magnitudes vary ($1.27 \pm 0.49$ TgC yr$^{-2}$, $0.28 \pm 0.11\,\%$ yr$^{-1}$). Specifically, VISIT-S3(G1997) projects the largest increase ($+1.79$ TgC yr$^{-2}$), followed by NorESM2-LM(G2012) and CESM2-WACCM(G2012) ($+1.71$ and $+1.54$ TgC yr$^{-2}$, respectively). GFDL-ESM4(G2006) and GISS-E2.1-G(G1995) show moderate increases ($+1.02$ and $+1.03$ TgC yr$^{-2}$, respectively), whereas UKESM1-0-LL(P2011) shows the smallest increase ($+0.52$ TgC yr$^{-2}$). In contrast, this period (1980–2014) showed a similar increase in global temperature for all models, of $+0.024$ to $+0.041\,°C$ yr$^{-1}$ ($p < 0.05$) among all models. The remarkable rise in temperature during this period can likely be attributed to the surge in GHG concentrations, particularly $CO_2$, and the decline in aerosol levels relative to the preceding period (1850–1979). It is particularly interesting that among models using the Guenther scheme, those with both $CO_2$ effects (CESM2-WACCM(G2012), NorESM2-LM(G2012)) and $CO_2$ fertilization only (VISIT-S3(G1997)) exhibit more positive trends in isoprene emissions than in models without $CO_2$ effects. This predominance of positive trends implies that, in addition to increasing temperature, $CO_2$ fertilization might be the second most important factor contributing to the marked increases in isoprene emission simulated in these models. For UKESM1-0-LL(P2011), even though its temperature trend is similar to other models, the smaller increase in isoprene emissions compared to those found using other Guenther-based models might be explained by its different scheme, which we investigate and discuss as presented hereinafter.

Figure 5 reveals remarkable similarities in the spatial distribution of isoprene emission trends during 1850–2014 between CESM2-WACCM(G2012) and NorESM2-LM(G2012). These similarities are likely attributable to their

**Table 3.** Global isoprene emission and temperature trends over three periods: 1850–2014, 1850–1979, and 1980–2014. Bold values indicate that a trend is significant, with $p < 0.05$. CE8

| Model | 1850–2014 | | 1850–1979 | | 1980–2014 | |
|---|---|---|---|---|---|---|
| | Isoprene emissions $(\text{TgC yr}^{-2})$ | Temperature $(°\text{C yr}^{-1})$ | Isoprene emissions $(\text{TgC yr}^{-2})$ | Temperature $(°\text{C yr}^{-1})$ | Isoprene emissions $(\text{TgC yr}^{-2})$ | Temperature $(°\text{C yr}^{-1})$ |
| VISIT-S3(G1997) | **+0.31** | **+0.005** | **+0.13** | **+0.002** | **+1.79** | **+0.024** |
| CESM2-WACCM(G2012) | +0.02 | **+0.005** | −0.11 | **+0.002** | **+1.54** | **+0.041** |
| NorESM2-LM(G2012) | +0.06 | **+0.002** | −0.10 | 0.000 | **+1.71** | **+0.037** |
| GFDL-ESM4(G2006) | **+0.13** | **+0.003** | +0.09 | **+0.002** | **+1.02** | **+0.031** |
| GISS-E2.1-G(G1995) | **+0.15** | **+0.005** | +0.07 | **+0.002** | **+1.03** | **+0.026** |
| UKESM1-0-LL(P2011) | **−0.92** | **+0.002** | **−0.92** | 0.000 | **+0.52** | **+0.034** |

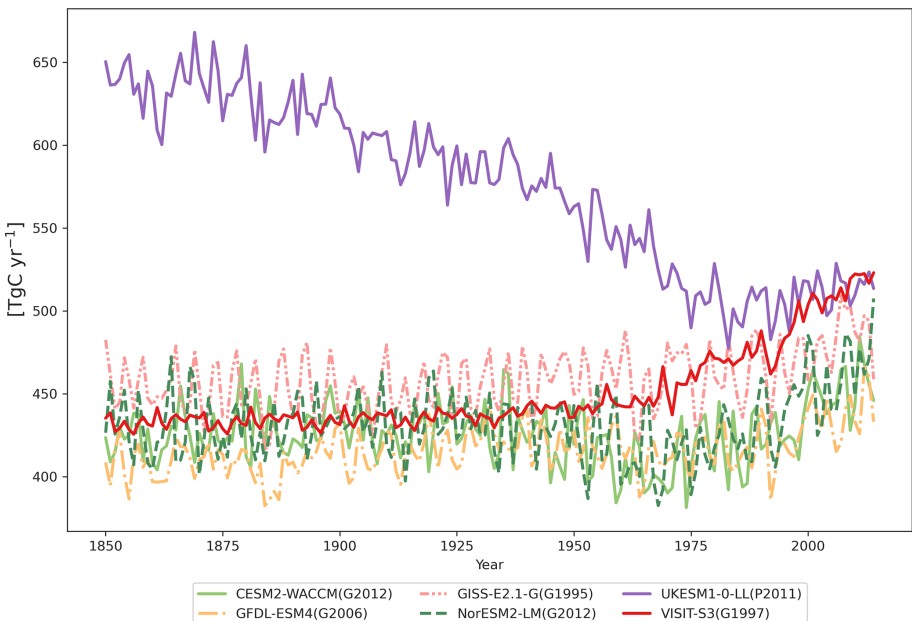

**Figure 4.** Interannual variations in global isoprene emissions during 1850–2014.

shared land model (CLM5), which incorporates the same version of the isoprene emission scheme G2012, as well as a shared atmosphere model (CAM6), albeit with somewhat different parameterizations and tuning. This pattern diverges from those of other models. Notably, all models employing the Guenther scheme agree on marked increases in specific regions such as the Amazon and central Africa, although these trends differ in terms of magnitude. Discrepancies emerge primarily in other regions. In South America and southern Africa, GISS-E2.1-G(G1995), VISIT-S3(G1997), and GFDL-ESM4(G2006) project a wider area of rising isoprene emission, whereas CESM2-WACCM(G2012), NorESM2-LM(G2012), and UKESM1-0-LL(P2011) suggest a larger area of decline. UKESM1-0-LL(P2011) exhibits a decrease in most regions, except in parts of western North America (WNA) and central Europe (CEU). Furthermore, high-latitude regions (north of 60° N) were found to have no significant trends in isoprene emis-

sions in CESM2-WACCM(G2012), NorESM2-LM(G2012), GFDL-ESM4(G2006), and GISS-E2.1-G(G1995). By contrast, UKESM1-0-LL(P2011) projects a considerable decrease, whereas VISIT-S3(G1997) shows an increase in these regions.

### 3.3 Contribution of drivers to global isoprene emission changes

Elevated $CO_2$, LULCC, and climate effects on annual global isoprene emission changes in individual models during 1850–2014 are shown in Fig. 6. Figure 7 presents a comparison of these trends during 1850–2014. The models including $CO_2$ effects indicated that most of the long-term trends in isoprene emissions can be attributed to $CO_2$ and LULCC. Four models consistently project a gradual decrease in emissions because of LULCC from 1850, although the magnitudes vary. CESM2-WACCM(G2012) shows the largest de-

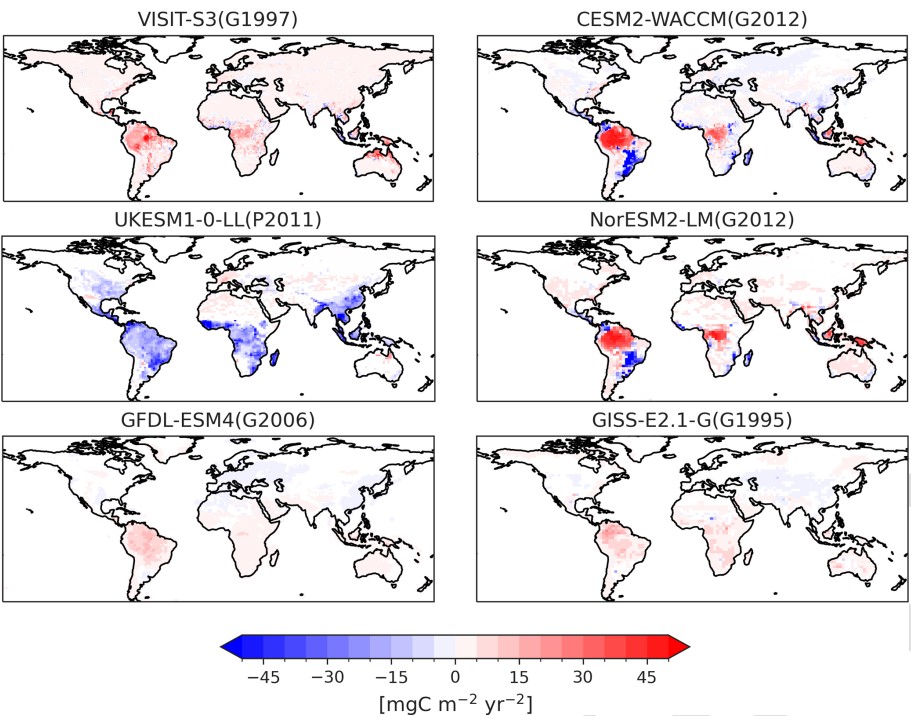

**Figure 5.** Spatial distribution of isoprene emission trends during 1850–2014. Only significant trends (with $p < 0.05$) are presented.

crease ($-0.42\,\mathrm{TgC\,yr^{-2}}$), followed closely by NorESM2-LM(G2012), UKESM1-0-LL(P2011), and VISIT(G1997) ($-0.27$, $-0.24$, and $-0.23\,\mathrm{TgC\,yr^{-2}}$, respectively). However, the $CO_2$ effect here, representing the net total effects of fertilization (increase) and inhibition (decrease), remains highly uncertain. While VISIT(G1997), CESM2-WACCM(G2012), and NorESM2-LM(G2012) show positive effects ($+0.42$, $+0.36$, and $+0.26\,\mathrm{TgC\,yr^{-2}}$, respectively), UKESM1-0-LL(P2011) exhibits a negative effect of $CO_2$ on the isoprene long-term emission trend ($-0.61\,\mathrm{TgC\,yr^{-2}}$). The underlying source of uncertainty related to these divergent trends among the models is discussed in Sect. 4.1.3. Although climate exerts a slight effect on long-term trends in isoprene emissions, it influences interannual variation within this model group. In contrast, GFDL-ESM4(G2006) and GISS-E2.1-G(G1995), excluding the $CO_2$ effects, attribute long-term emission trends primarily to climate ($+0.12$ and $+0.15\,\mathrm{TgC\,yr^{-2}}$, respectively), with LULCC playing a negligible role ($-0.0005\,\mathrm{TgC\,yr^{-2}}$).

Figures 8 and 9 present individual climate factors influencing the isoprene emission trends. Climate is a strong driver of year-to-year variation in isoprene emissions in all models, with the most decisive influence in GFDL-ESM4(G2006) and GISS-E2.1-G(G1995). Empirically based CMIP6 models agree that temperature is the primary driver of interannual variation in isoprene emissions, whereas the photosynthesis-based model UKESM1-0-LL(P2011) highlights radiation and precipitation and VISIT(G1997) emphasizes temperature and precipitation over radiation (Fig. 8). Interan-

nual variations in these climate variables primarily drive interannual changes in isoprene emissions across models. VISIT(G1997) notably exhibits lower interannual variation in isoprene emissions than that in the CMIP6 models, primarily because it relies on reanalysis climate data, which contain less variability compared to the climate data simulated by the CMIP6 models (Fig. S5 and Table S5). The small interannual variability during 1850–1900 simulated by VISIT could come from the spin-up method using climatological forcing data, with the model being repeatedly driven by the climate data of 1901 during this period. Furthermore, although GISS-E2.1-G(G1995) uses a temperature-dependent algorithm similar to that of VISIT(G1997) (Table S1), the greater temperature variability in GISS-E2.1-G(G1995) results in larger interannual variation in isoprene emissions. Regarding long-term trends, climate factors exert a stronger influence on emission trends in GFDL-ESM4(G2006) and GISS-E2.1-G(G1995) compared to the other models. All models show agreement on temperature-increasing isoprene emission, but the specific effects vary: $+0.04$ to $+0.18\,\mathrm{TgC\,yr^{-2}}$. However, the VISIT(G1997) model reveals that radiation causes only a minor increase ($+0.005\,\mathrm{TgC\,yr^{-2}}$, $p > 0.05$). In contrast, CMIP6 models indicate that decreased radiation contributes to a minor reduction in isoprene emission, ranging from $-0.02$ to $-0.04\,\mathrm{TgC\,yr^{-2}}$ ($p < 0.05$). The effects of precipitation further highlight discrepancies among models: VISIT(G1997) shows an increase in emissions ($+0.02\,\mathrm{TgC\,yr^{-2}}$), whereas UKESM1-0-LL(P2011) projects a decrease ($-0.04\,\mathrm{TgC\,yr^{-2}}$), and other models show non-

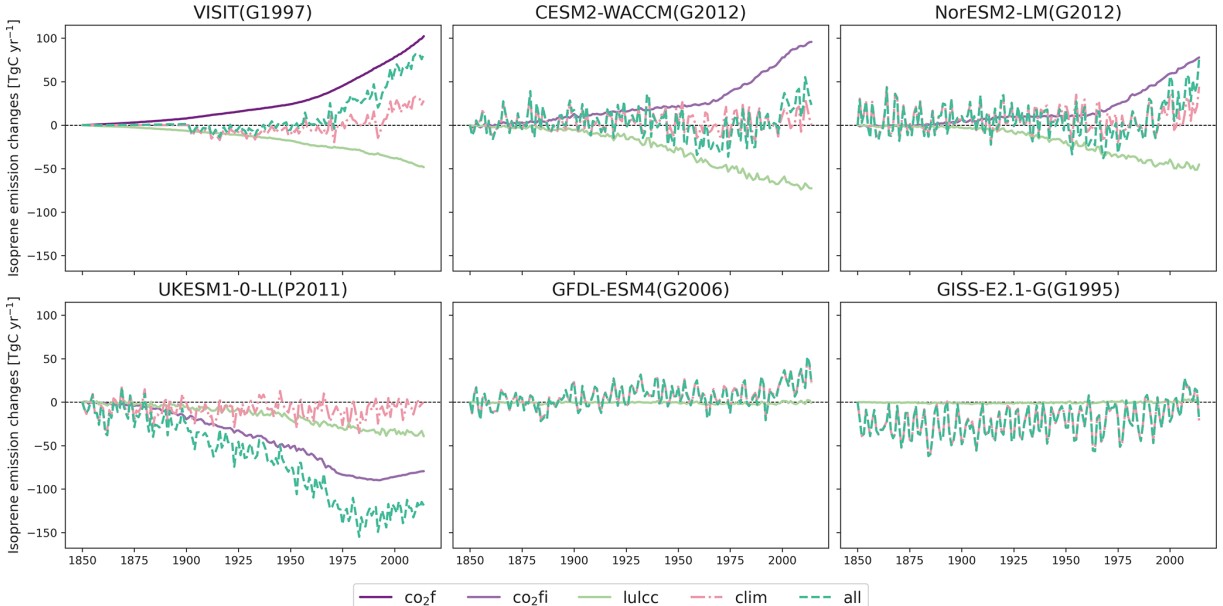

**Figure 6.** Effects of drivers on global isoprene emissions during 1850–2014: $CO_2$ (co$_2$f, $CO_2$ fertilization only; co$_2$fi, combined $CO_2$ fertilization and inhibition), land use and land cover change (lulcc), climate (clim), and the combination of three drivers (all). These effects were calculated as the difference in isoprene emission simulations between fixed and varied states for each driver, details of which are described in Sect. 2.3.1.

significant trends ($+0.003$ to $+0.006$ TgC yr$^{-2}$) attributed to precipitation.

### 3.4 Spatial contribution of drivers when estimating isoprene emissions

Figure 10 shows that $CO_2$ and LULCC influence isoprene emission changes more than climate in the models including the $CO_2$ effect. $CO_2$ primarily drives increased emissions, especially in tropical regions, for VISIT(G1997), CESM2-WACCM(G2012), and NorESM2-LM(G2012), whereas UKESM1-0-LL(P2011) exhibits a decreasing trend. However, these models show agreement in that LULCC engenders decreasing emissions, particularly in regions such as central Africa and southern Africa, as well as South Asia and Southeast Asia. However, parts of Europe and eastern North America show increases. In southeastern South America (SSA), CESM2-WACCM(G2012), NorESM2-LM(G2012), and UKESM1-0-LL(P2011) (excluding VISIT(G1997)) identify LULCC (deforestation) as the main driver of emission changes. However, GFDL-ESM4(G2006) and GISS-E2.1-G(G1995), without $CO_2$ effects, find the LULCC effect on isoprene emission changes to be minimal compared to the other models. Regarding climate effects on isoprene emission, differences in climate variability explain the discrepancies in model contributions.

A spatial distribution of the most dominant variables influencing isoprene emission trends across $CO_2$, LULCC, and climate is portrayed in Fig. 11. Overall, $CO_2$ emerges as the dominant driver for 34 %–63 % of global land area in VISIT(G1997), CESM2-WACCM(G2012), NorESM2-LM(G2012), and UKESM1-0-LL(P2011). Here, LULCC plays a minor role, accounting for 14 %–25 % of the global land area in all models, but its spatial distribution of dominance varies between them. Effects of climate further add to inter-model variation. While UKESM1-0-LL(P2011), CESM2-WACCM(G2012), and NorESM2-LM(G2012) show climate affecting 23 %–32 % of global land area, VISIT(G1997) presents a much more extensive effect, with climate dominating 46 % of land. By contrast, GFDL-ESM4(G2006) and GISS-E2.1-G(G1995), without the $CO_2$ effect, reveal climate-related factors as the most dominant drivers, affecting 82 % and 75 %, respectively, of the global land area.

Regionally, VISIT(G1997), CESM2-WACCM(G2012), NorESM2-LM(G2012), and UKESM1-0-LL(P2011) identify $CO_2$ effects as the primary driver of isoprene emissions in tropical regions such as the Amazon and central Africa (Fig. 11). Whereas $CO_2$ increases emissions in the first three models, it engenders a decrease in UKESM1-0-LL(P2011). LULCC emerges as the dominant driver across all models despite variations in emission schemes and climate factors in Southeast Asia and Europe. West Africa and East Africa show a more complex picture: VISIT(G1997), CESM2-WACCM(G2012), and NorESM2-LM(G2012) identify LULCC in addition to $CO_2$ as the primary drivers there, while UKESM1-0-LL(P2011) points to $CO_2$ and GFDL-ESM4(G2006) and GISS-E2.1-

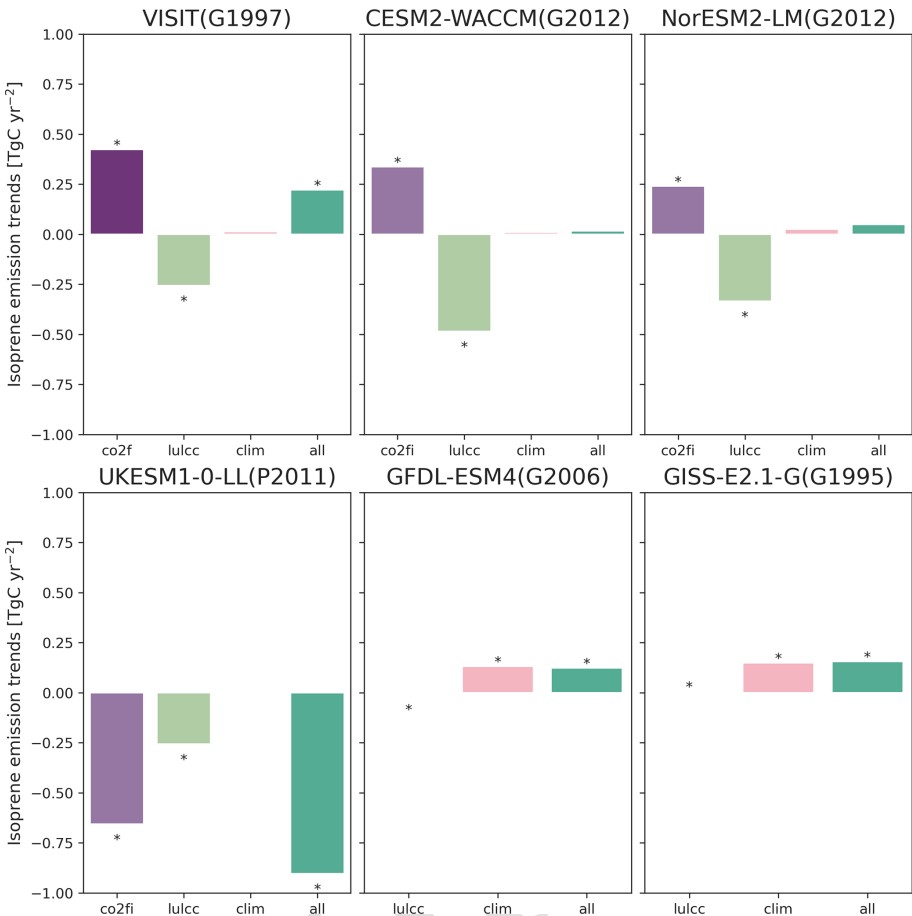

**Figure 7.** Attribution of global isoprene emission trends during 1850–2014 attributable to each driver ($co_2f$/$co_2fi$, lulcc, and clim) and the combination of three drivers (all). Asterisks denote a significant trend, with $p < 0.05$.

G(G1995) point to climate. Southeastern South America also presents discrepancies: CESM-WACCM(G2012), NorESM2-LM(G2012), and UKESM1-0-LL(P2011) indicate LULCC as the main driver, while VISIT(G1997) and the other two models indicate climate factors.

Figure 12 highlights the distinct contributions of individual climate factors to long-term trends in isoprene emissions across models between 1850 and 2014. These differences reflect the varying spatial distributions of the climate variables themselves. Temperature and radiation stand out for their large contributions, while precipitation plays a minor role in most CMIP6 models. Temperature's effects reach a peak in tropical regions, while radiation's influence is greatest in the tropics and some middle and high northern latitudes in these models. VISIT(G1997) stands alone, with both temperature and precipitation exerting stronger effects than radiation. This finding might be attributable to VISIT's big-leaf canopy model, which is less responsive than other models to changes in radiation. The effect of precipitation is particularly pronounced in the tropics, whereas temperature plays a leading role in southeastern South America

and East Africa. Unsurprisingly, one key point of agreement across models is that temperature increases generally engender higher isoprene emissions across most regions, reflecting the well-established relation captured by most models. Conversely, surface radiation typically engenders decreases in most models, with VISIT(G1997) again being the exception. Precipitation-driven changes remain highly uncertain, in both magnitude and sign of the trends. For example, VISIT(G1997) shows marked increases in Amazonia emissions because of increased precipitation, whereas UKESM1-0-LL(P2011) projects a decrease in the same region because of reduced precipitation. This discrepancy results from the different climate datasets: VISIT(G1997) relies on reanalysis data, while UKESM1-0-LL(P2011), like other CMIP6 models, uses its own modelled climate data (Fig. S7).

Figure 13 paints a contrasting picture of dominant climate drivers for isoprene emissions across models. Temperature reigns supreme across most regions for models using the Guenther scheme, affecting emission changes in 59 %, 69 %, and 73 % of the global area in GFDL-ESM4(G2006), VISIT(G1997), and GISS-E2.1-G(G1995),

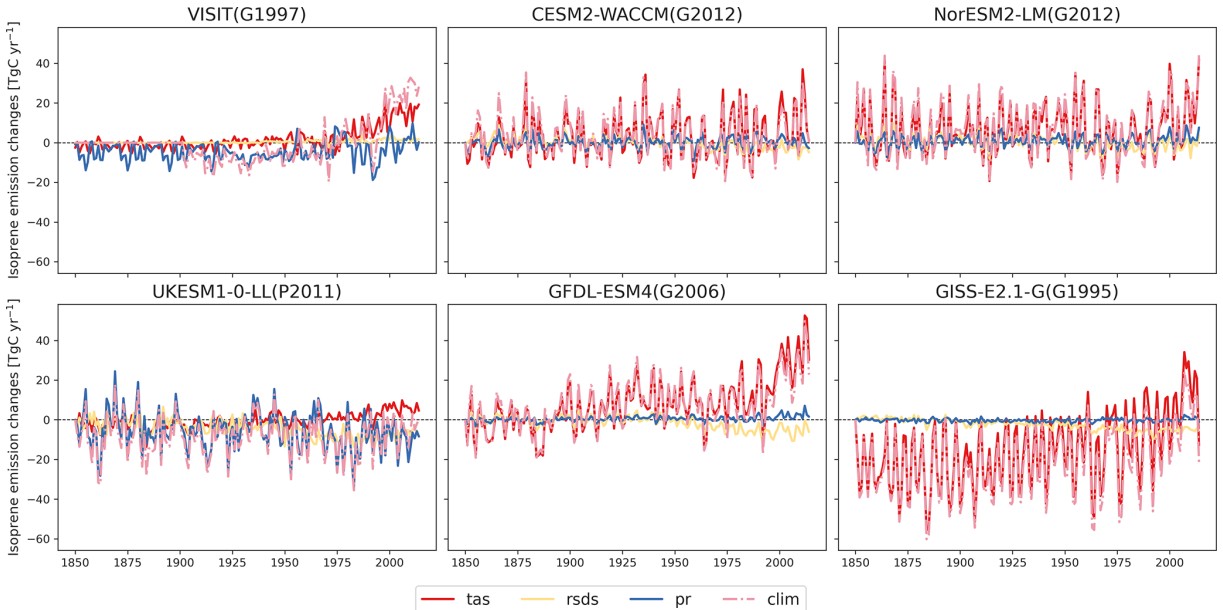

**Figure 8.** Effects of climate factors on global isoprene emissions during 1850–2014: temperature (tas), shortwave radiation (rsds), precipitation (pr), and total climate factors (clim).

respectively. This figure greatly exceeds the 40 %–46 % influence observed for other CMIP6 models. By contrast, UKESM1-0-LL(P2011) stands out, with radiation leading in nearly half of the global land area (45 %), compared to the 24 %–38 % range for other CMIP6 models. VISIT(G1997) stands apart from CMIP6 models by demonstrating a limited effect of radiation on emission changes, affecting only 7 % of the global land area. Precipitation's effects on isoprene emission trends vary markedly among models. GISS-E2-1-G(G1995) and GFDL-ESM4(G2006) show minimal effects, with precipitation dominating only 3.5 % and 8 %, respectively, of global land area. Other models exhibit moderate effects, with precipitation affecting 20 %–25 % of global land area.

On a regional scale, the dominant meteorological driver of isoprene emission changes differed substantially among models. In the Amazon, VISIT(G1997) identifies precipitation as the primary driver, whereas the other models point to temperature. Central Africa and Southeast Asia show similar patterns, with temperature dominating in all models except UKESM1-0-LL(P2011), in which radiation and precipitation jointly exert influences. In mid-latitude and high-latitude northern regions, radiation leads across models, although VISIT(G1997) shows a weaker effect. All models show agreement on the considerable influence of precipitation in specific arid and semiarid regions such as the Sahara and South Asia. In Australia, temperature dominates in GFDL-ESM4(G2006) and GISS-E2.1-G(G1995), while VISIT(G1997) identifies precipitation as the dominant factor. However, UKESM1-0-LL(P2011) shows no significant

trend in isoprene emissions attributable to any meteorological factor in this region.

Figure 14 shows the inter-model spread, representing the standard deviation of isoprene emission trends attributable to each factor, calculated for each grid cell across all models. The inter-model spreads of isoprene emission trends are located primarily in tropical areas and the Southern Hemisphere (Fig. 14). The highest uncertainty in the contribution of $CO_2$ to isoprene emission trends in tropical areas (Fig. 14a) arises from inconsistencies in $CO_2$ effects among models. $CO_2$ consistently shows a positive influence on isoprene emissions in the tropics in VISIT(G1997), CESM2-WACCM(G2012), and NorESM2-LM(G2012), but it has a negative effect in UKESM1-0-LL(P2011) (Fig. 10a). Also, uncertainties in LULCC effects on isoprene trends are most concentrated in southeastern South America and Southeast Asia (Fig. 14b). The reason underlying uncertainties in LULCC effects might arise from variations in original land cover maps, land use schemes, and emission factors across models. For instance, in southeastern South America, CESM2-WACCM(G2012) and UKESM1-0-LL(P2011) simulate a stronger effect of LULCC because of that region's conversion of forest to cropland compared to GFDL-ESM4(G2006), which shows more minor changes in isoprene emissions, mainly because of its conversion of grassland to cropland, as shown in Fig. S4. In contrast, inter-model differences driven by climate factors are more minor than those for $CO_2$ and LULCC (Fig. 14c). This is true because the inhomogeneous effects of different climate elements on isoprene emissions tend to offset each other to a large degree (Figs. 10c and 12). Among the climate variables, uncer-

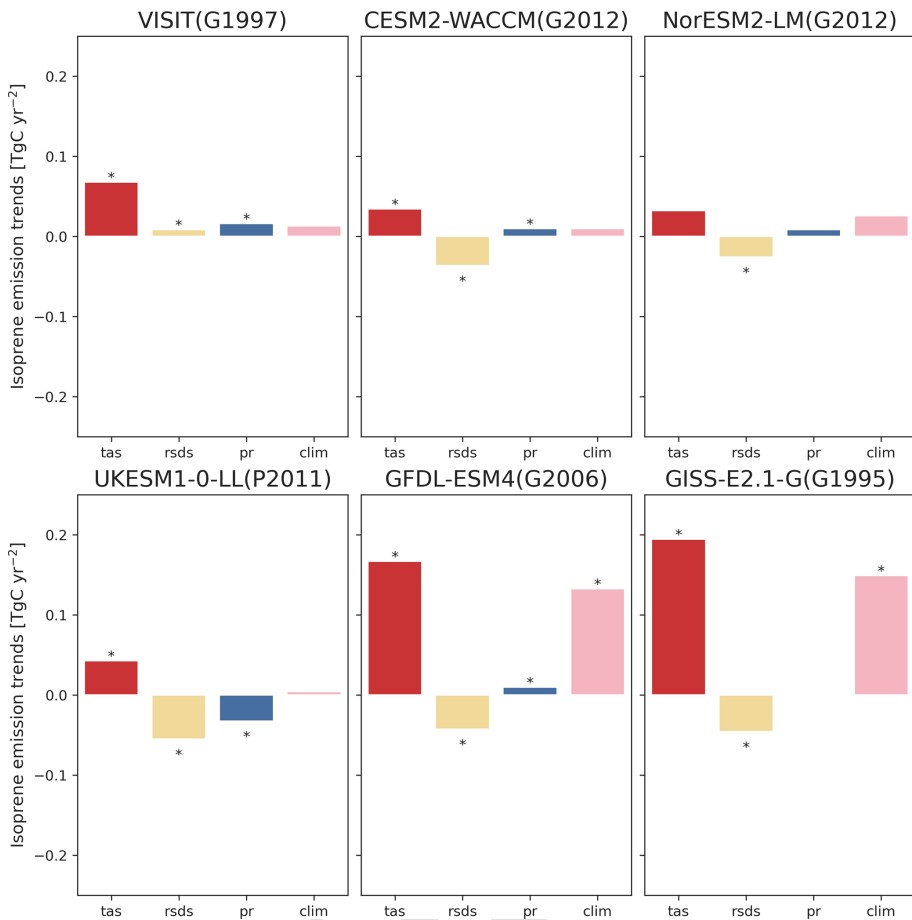

**Figure 9.** Attribution of global isoprene emission trends during 1850–2014 attributable to each climate factor (tas, rsds, pr) and total climate effects (clim). Asterisks denote a significant trend, with $p < 0.05$.

tainties in the effects of temperature on isoprene trends are concentrated in the tropics, whereas those for radiation are concentrated in middle northern latitudes (e.g. central North America and East Asia) and those for precipitation in parts of the Amazon, Madagascar, and northern Australia.

## 4 Discussion and perspective

### 4.1 Sources of uncertainty

#### 4.1.1 Variability in regional isoprene emissions

Models show remarkable consistency in estimation of the global quantities of isoprene emissions for recent times, but considerable regional discrepancies exist (Fig. 1b). Tropical regions, notably the Amazon ($98–175\,\mathrm{TgC\,yr^{-1}}$) and Southeast Asia ($14.7–87.8\,\mathrm{TgC\,yr^{-1}}$), exhibit the greatest variability. Arid regions such as the Sahara also displayed wide ranges ($0.1–24.9\,\mathrm{TgC\,yr^{-1}}$) (Table S4). The main reason underlying these regional discrepancies appears to be differences in how models represent plant functional types

(PFTs) and emission factors assigned to each PFT across the models (Tables S6 and S7). For instance, the latest Guenther scheme (G2012) used in CESM2-WACCM(G2012) and NorESM2-LM(G2012) incorporated 16 PFTs, compared to 7 or 11 PFTs in the older versions used by GFDL-ESM4(G2006) or GISS-E2.1-G(G1995) and 16 PFTs for VISIT(G1997). For the P2011 scheme, 13 PFTs were used for UKESM1-0-LL(P2011). Moreover, the definitions of these PFTs vary among models, influencing their emission factors (Table S7). For instance, in the G2012 scheme, emission factors for broadleaf trees (evergreen vs. deciduous) across tropical, temperate, and boreal regions range from 20.6 to $52.4\,\mathrm{\mu gC\,g_{mass}^{-1}\,h^{-1}}$, with the highest emissions from broadleaf deciduous boreal trees. Also, VISIT(G1997) assigned emission factors of $8–45\,\mathrm{\mu gC\,g_{mass}^{-1}\,h^{-1}}$ to five broadleaf trees, but the highest emission factor was assigned to broadleaf deciduous temperate trees. The emission factor for a single broadleaf tree type in the G2006 scheme is only $24\,\mathrm{\mu gC\,g_{mass}^{-1}\,h^{-1}}$, whereas the G1995 scheme assigns $24\,\mathrm{\mu gC\,g_{mass}^{-1}\,h^{-1}}$ for broadleaf evergreen trees and $24/45\,\mathrm{\mu gC\,g_{mass}^{-1}\,h^{-1}}$ for two broadleaf deciduous trees de-

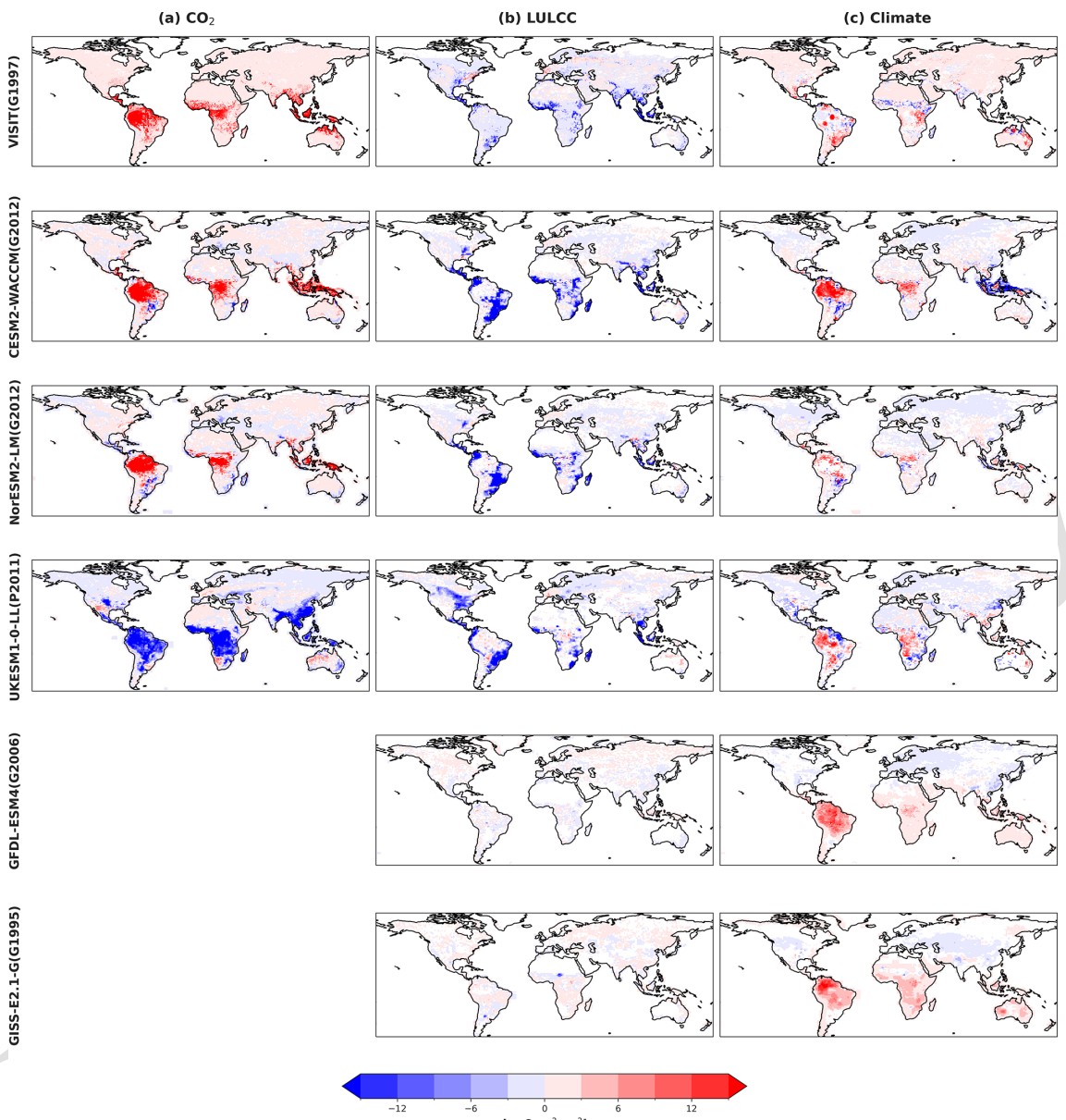

**Figure 10.** Spatial distributions of the contributions of **(a)** $CO_2$, **(b)** LULCC, and **(c)** climate to the isoprene emission trends in the respective models. Only significant trends (with $p < 0.05$) are presented. It is noteworthy that GFDL-ESM4(G2006) and GISS-E2.1-G(G1995) do not include the $CO_2$ effects.

pending on whether they are cold- or drought-tolerant. The P2011 scheme uses emission factors ranging from 16–35 $\mu gC\ g_{mass}^{-1}\ h^{-1}$ assigned for three broadleaf tree types, with the highest emission for broadleaf deciduous trees across all regions.

These variations in PFT representation and emission factors strongly influence the spatial distributions of isoprene emissions among models. For instance, in the Amazon, G2012 assigned a tree fraction of roughly $74 \pm 0.13\ \%$ between 2000 and 2014, while G2006 and G1995 allocated only $34 \pm 0.23\ \%$. By contrast, G2006 estimated a larger grass fraction ($36 \pm 2.50\ \%$) compared to G2012 ($21 \pm 0.14\ \%$) (Fig. S4). Although these models did not provide specific information related to tree types, broadleaf evergreen trees are generally predominant in this region, emitting more isoprene than grasses (Table S7). This greater emission explains why the higher tree fraction of G2012 caused markedly higher total isoprene emissions for the Amazon (175 TgC yr$^{-1}$) than the average isoprene emissions in G2006 and G1995 (103 TgC yr$^{-1}$). Also, VISIT(G1997) and P2011, with similar emission factors for broadleaf evergreen trees and $C_4$ grass (24 $\mu gC\ g_{mass}^{-1}\ h^{-1}$), exhibited comparable

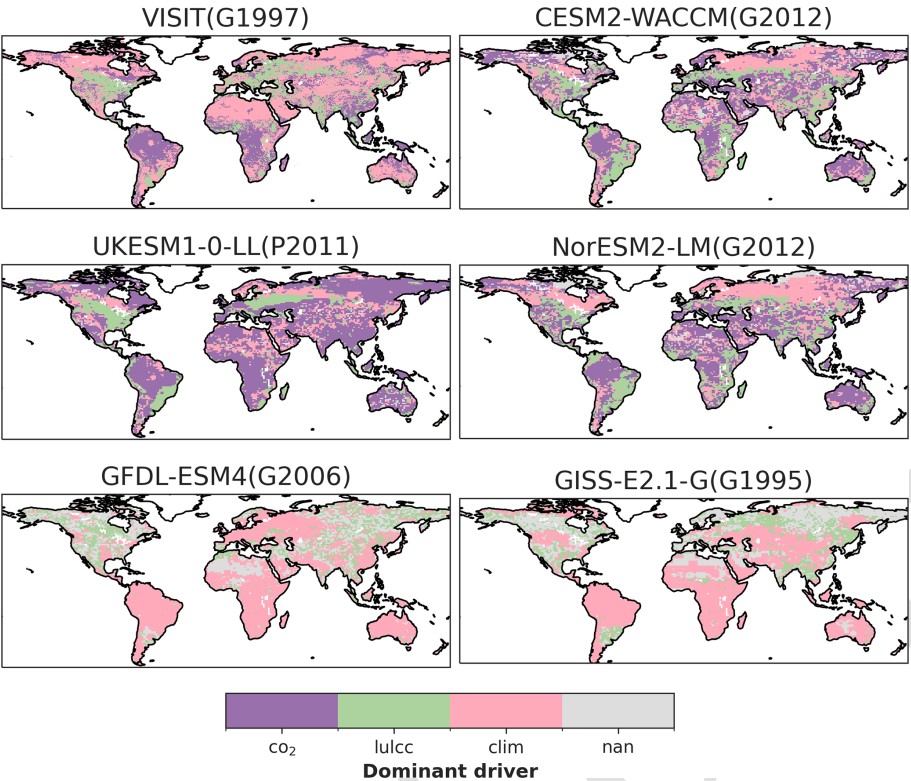

**Figure 11.** Dominant driver of isoprene emission trends between 1850 and 2014. For each grid, the factor generating the absolute largest trend is selected as the dominant driver. "nan" denotes no significant trend in isoprene emissions because of any factor. It is noteworthy that GFDL-ESM4(G2006) and GISS-E2.1-G(G1995) do not include the $CO_2$ effects.

emissions in this region (133 vs. $117\,\mathrm{TgC\,yr^{-1}}$). P2011 allocated a tree fraction of approximately $75 \pm 0.20\,\%$, similar to that of G2012, and a grass fraction of $11.7 \pm 0.13\,\%$, half that of G2012. The emission factor for broadleaf evergreen trees in G2012 ($20.6\,\mathrm{\mu gC\,g_{mass}^{-1}\,h^{-1}}$) is slightly lower than that in P2011 ($24\,\mathrm{\mu gC\,g_{mass}^{-1}\,h^{-1}}$), but the emission factor for $C_4$ grass in P2011 ($24\,\mathrm{\mu gC\,g_{mass}^{-1}\,h^{-1}}$) is notably 20 times higher than in G2012 ($1.2\,\mathrm{\mu gC\,g_{mass}^{-1}\,h^{-1}}$), which was identified as a bug in P2011 (Weber et al., 2023) and which was corrected in current development strands of UKESM1-0-LL. However, compensation effects between the two plant types cannot fully explain the lower isoprene emissions in UKESM1-0-LL(P2011) than CESM2-WACCM(G2012) over the Amazon. Another reason might be the difference in the isoprene emission scheme between these models, such as the scaling factor to adjust for variations in standard temperature ($297\,\mathrm{K}$ in the Guenther scheme vs. $303.15\,\mathrm{K}$ in P2011) for setting basal emission factors. This hypothesis, suggesting that emission factors strongly influence the spatial distribution of isoprene emissions and regional quantities of isoprene emissions, is further supported by reductions observed when using different emission factors in VISIT(G1997) (Ito, 2019b). Applying the standard high emission factor ($24\,\mathrm{\mu gC\,g_{mass}^{-1}\,h^{-1}}$) yielded global isoprene emissions of $510\,\mathrm{TgC\,yr^{-1}}$, while the lower emission factor ($9\,\mathrm{\mu C\,\mu gC\,g_{mass}^{-1}\,h^{-1}}$), based on Malaysian observations for broadleaf evergreen forests (Saito et al., 2008), reduced mean global emissions to $342\,\mathrm{TgC\,yr^{-1}}$ (33 %) during 2000–2014, with Amazonia emissions dropping from 133 to $62\,\mathrm{TgC\,yr^{-1}}$ (54 %). Similarly, the stronger isoprene emissions simulated by the GISS-E2.1-G(G1995) in northern Australia and the Sahara likely arise from a higher proportion of shrubs with higher emission factors in its vegetation representation compared to other models, which grasses might dominate (Figs. S4 and S8). However, changes in PFTs and their associated emission factors primarily influence the spatial distribution of isoprene emission, not the seasonality, in VISIT(G1997) and other models employing MEGAN or P2011 schemes (Henrot et al., 2017; Weber et al., 2023). Unfortunately, assessing details of the uncertainty arising from PFTs and their differences in emission factors is beyond the scope of this study because of the lack of necessary output data available from the CMIP6 models. However, future efforts to establish a standardized global PFT map with corresponding PFT-specific emission factors hold great promise for reducing these uncertainties and for improving the consistency of simulations across different models.

Current approaches that use generalized PFT emission factors may oversimplify the effects of climate-driven shifts in vegetation composition. Changes in tree species distribu-

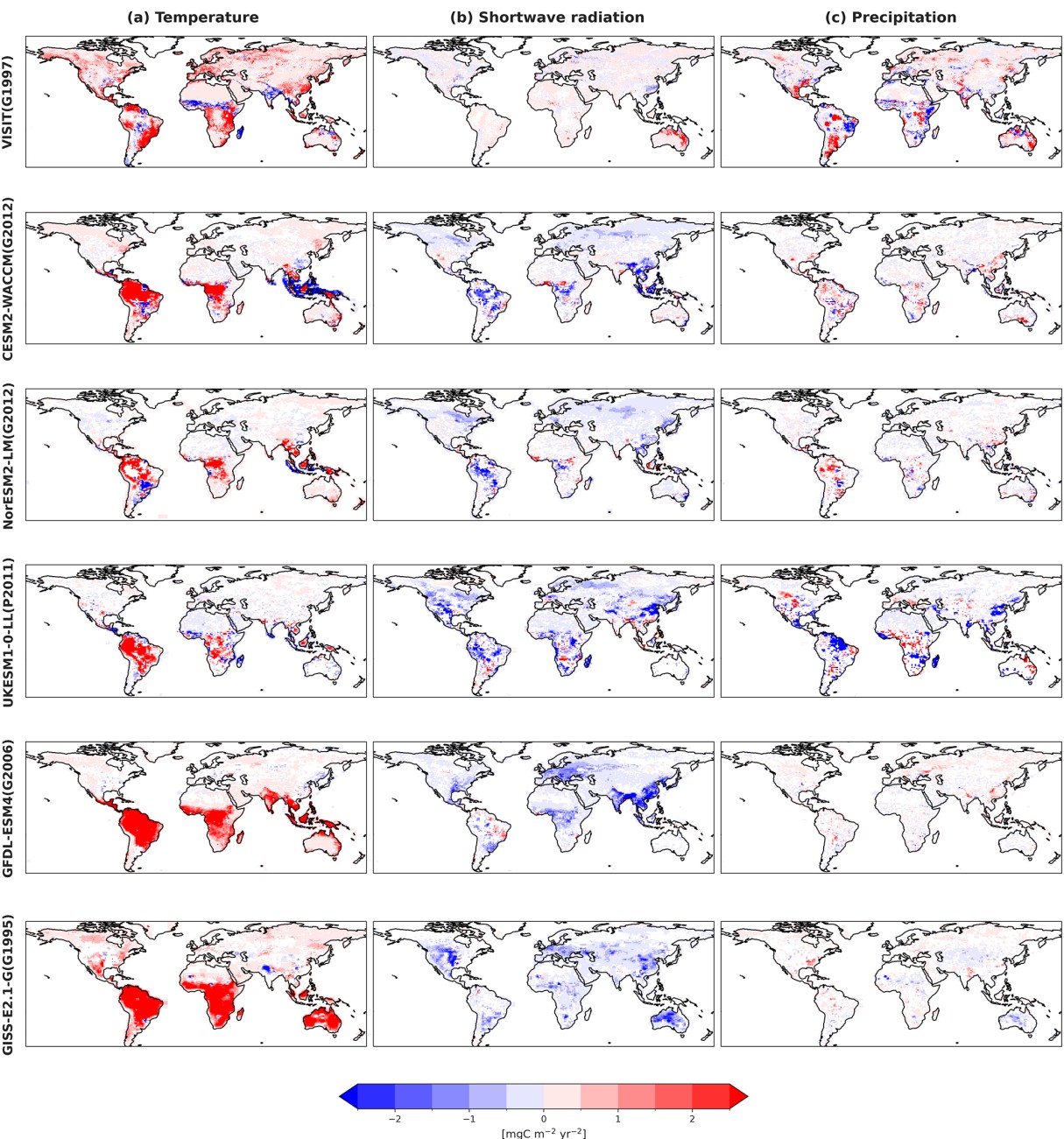

**Figure 12.** Spatial distribution of the contribution of each climate factor – **(a)** temperature, **(b)** shortwave radiation, and **(c)** precipitation – to the isoprene emission trends in each model. Only significant trends (with $p < 0.05$) are presented.

tion and composition in response to climate change affect the quantity and composition of BVOC emissions. Isoprene emission rates vary widely across plant species, and assigning emission factors to specific PFTs is often not unambiguous. Plant-specific emission factors would more accurately represent the impacts of individual climate drivers, as emission factors can differ greatly even within the same genus (Karl et al., 2009; Satake et al., 2024). Dani et al. (2014) suggested that the trait of isoprenoid emission in evergreen plants can be lost during evolution in favour of more storable compounds (monoterpenes) to better cope with repeated and prolonged stress. Additionally, the proportion of isoprene-emitting tropical trees appears to increase with mean annual temperature but decrease with the length of the dry season (Taylor et al., 2018). These findings suggest that future model developments should also consider plant-specific emission factors and phenological changes to accurately assess the im-

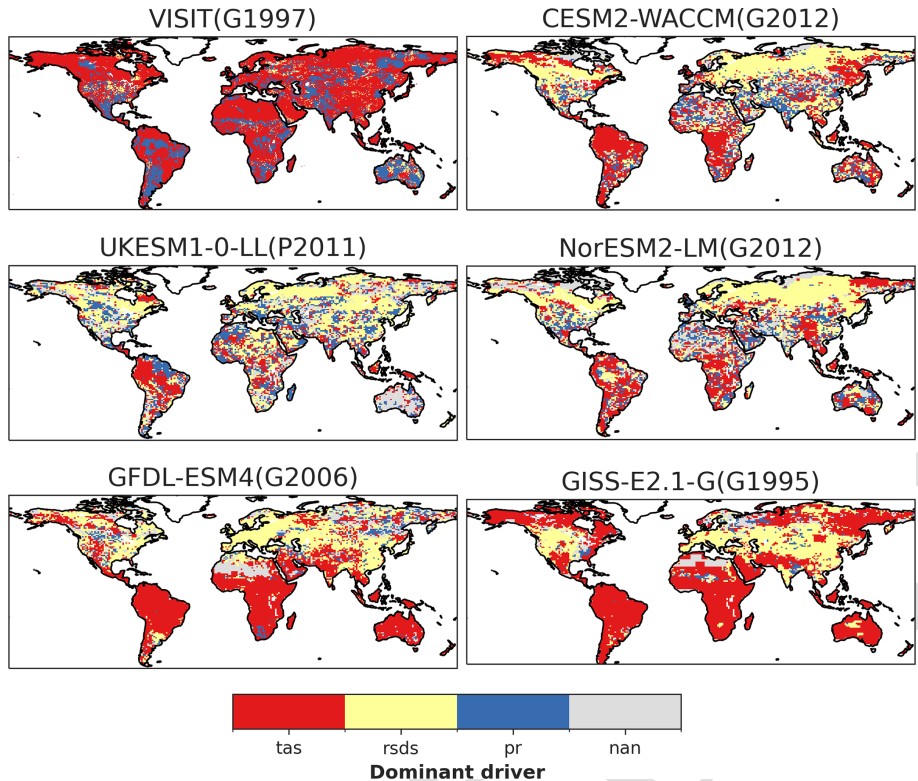

**Figure 13.** Dominant meteorological drivers of isoprene emission trends during 1850–2014. For each grid, the factor generating the absolute largest trend is selected as the dominant driver. "nan" denotes no significant trend in isoprene emissions attributable to any factor.

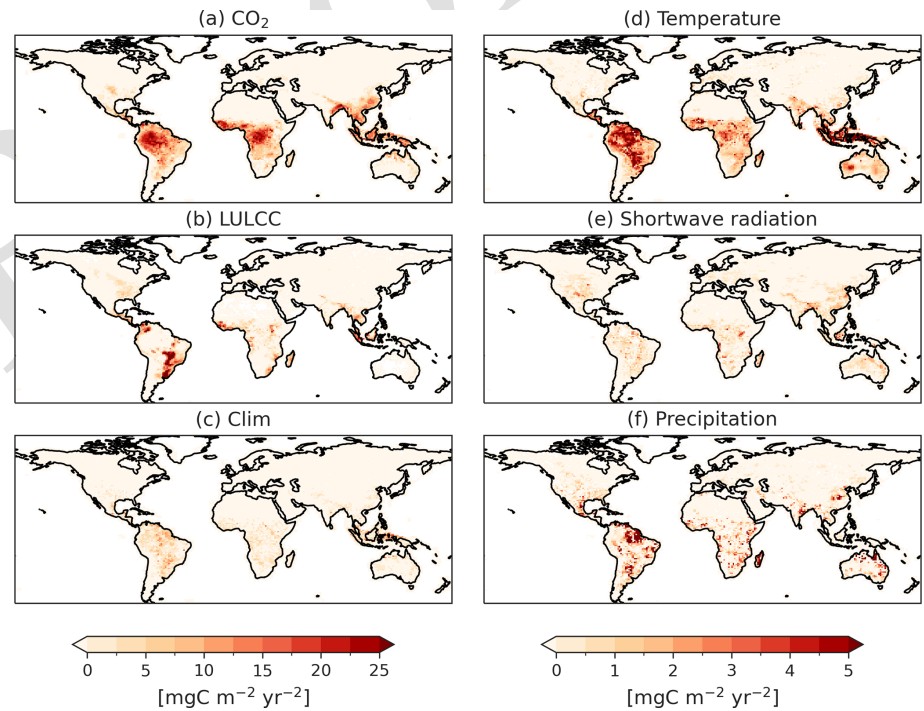

**Figure 14.** Inter-model spreads of isoprene emission trends attributable to the following: **(a)** $CO_2$; **(b)** LULCC; **(c)** combined effects of three climate factors; and **(d–f)** the respective climate factors of temperature, shortwave radiation, and precipitation.

pacts of climate-driven changes in tree species distribution and composition on BVOC emissions.

### 4.1.2 Variability in attribution to isoprene emissions

In recent decades (1980–2014), all the models detected an overall increasing trend in global isoprene emissions, which agrees with findings from earlier studies using satellite data with MEGAN (Opacka et al., 2021). However, the main drivers of this trend differed among the models (VISIT and the CMIP6 models) (Fig. 6) depending on the assumptions considered in the parameterization (Table S1). These models agree that the $CO_2$ fertilization effect is dominant over the $CO_2$ inhibition effect in MEGAN-based models for this period. Alternatively, there are compensatory effects between $CO_2$ inhibition and temperature in P2011 (Pacifico et al., 2012). This finding in MEGAN-based models in our study aligns with those findings reported by Heald et al. (2009), who claimed that while $CO_2$ inhibition partially offsets the effect of rising temperatures on isoprene emissions, it does not fully compensate for the total effects of rising temperature and vegetation productivity. This inadequate compensation underscores the important role of $CO_2$ fertilization in future isoprene emission changes, as well as in trends observed for recent decades (1980–2014) in our study, which are partially reflected in the future snapshot simulation (2100) in the earlier study (Heald et al., 2009). Further investigation into isoprene emission trends and their controlling factors in future simulations (2015–2100) under CMIP6 ScenarioMIP (SSP) is necessary to validate the robustness of these findings. The magnitude of the net $CO_2$ effect remains highly uncertain, depending on the model scheme and how it accounts for $CO_2$ inhibition, if at all. Additionally, LULCC had a moderate effect on the emission trends in these models. By contrast, models without $CO_2$ effects (GFDL-ESM4(G2006) and GISS-E2.1-G(G1995)) showed a minimal influence of LULCC on the emission trends, whereas climate effects dominated, leading to lower increasing trends compared to other MEGAN-based models.

Regarding long-term trends (1850–2014), CESM2-WACCM(G2012) and NorESM2-LM(G2012) showed minimal changes in isoprene emissions (Figs. 6 and 7) because the $CO_2$ effects balanced out the LULCC effects. VISIT(G1997) exhibited trends that were more positive than those of either CESM2-WACCM(G2012) or NorESM2-LM(G2012) because of its inclusion of $CO_2$ fertilization but exclusion of $CO_2$ inhibition. Even with $CO_2$ inhibition included, CESM2-WACCM(G2012) and NorESM2-LM(G2012) showed long-term trends for isoprene emissions driven by $CO_2$ similar to those of VISIT(G1997) (Fig. 6). This similarity of trends suggests that $CO_2$ fertilization predominates over $CO_2$ inhibition, which might only become active when $CO_2$ concentrations exceed a threshold (e.g. 365 ppm in these models). The average isoprene emissions attributed to $CO_2$ in CESM2-WACCM(G2012) and NorESM2-LM(G2012) during 2000–2014 (1.259 TgC yr$^{-2}$) were lower than the average in VISIT(G1997) (1.714 TgC yr$^{-2}$). UKESM1-0-LL(P2011) exhibited a significant and negative trend in isoprene emissions because of the combined effects of $CO_2$ and LULCC. This singularity of UKESM1-0-LL(P2011) might be attributable to its different isoprene emission scheme (P2011), particularly its methodology to treat $CO_2$ effects (direct inhibition and indirect fertilization via photosynthesis (GPP) parameterization) compared to MEGAN-based models. Despite radiation-induced decreases, GFDL-ESM4(G2006) and GISS-E2.1-G(G1995) displayed increasing trends driven by rising temperatures. Among the three drivers ($CO_2$, LULCC, and climate), the $CO_2$ effects on isoprene emissions have the highest inter-model variability ($\sigma = 0.43$ TgC yr$^{-2}$), followed by LULCC ($\sigma = 0.17$ TgC yr$^{-2}$) and climate change ($\sigma = 0.06$ TgC yr$^{-2}$). Therefore, the different mechanisms used for the respective models to account for $CO_2$, LULCC, and climate effects contribute to the uncertainty in long-term global isoprene emission trends. Particularly addressing $CO_2$ and LULCC effects, rather than meteorological factors alone, is crucially important for the use of long-term models. All the models show agreement in terms of a significant increase from 1980 to the present, but pre-1980 trends remain uncertain, highlighting the need for improved historical data and refined model evaluation to capture past trends better and to enhance the accuracy of future predictions.

Regarding spatial distribution, the models without $CO_2$ effects (GFDL-ESM4(G2006) and GISS-E2.1-G(G1995)) generally predict a weaker gradient in isoprene emissions compared to other models (Fig. 5). This difference likely arises because isoprene emissions are not tied interactively to vegetation production in these two models. Higher $CO_2$ can either directly reduce (inhibition) or indirectly increase isoprene emissions because of enhanced vegetation production (represented by LAI in MEGAN or GPP in P2011). It is true that CESM2-WACCM(G2012), NorESM2-LM(G2012), UKESM1-0-LL(P2011), and VISIT(G1997) captured the indirect $CO_2$ effects, but VISIT(G1997) neglected the direct inhibition effect. Nevertheless, both effects were considered in CESM2-WACCM(G2012), NorESM2-LM(G2012), and UKESM1-0-LL(P2011). As the dominant driver in tropical regions (Fig. 11), $CO_2$ effects strongly influence the global isoprene emission trends. These regions are high emitters because of sustained warm temperatures, intense radiation, and high biomass density, coupled with high emission factors of tropical vegetation types (Henrot et al., 2017). The inter-model spread of $CO_2$'s contribution to isoprene emission trends is particularly pronounced in tropical areas, surpassing the spread associated with LULCC and climate effects (Fig. 14). This suggests that further model refinement is needed, particularly in accurately representing $CO_2$ effects on isoprene emissions in tropical regions.

Compared to temperature and radiation, precipitation is found to play a minor role in controlling long-term trends of

isoprene emissions in most CMIP6 models. However, the effect of extreme climate events, such as drought stress, on isoprene emissions is uncertain because of limitations in the algorithms representing drought impacts. While UKESM1-0-LL(P2011) and VISIT(G1997) account for indirect drought effects through photosynthesis, only NorESM2-LM(G2012) models direct effects using its own soil moisture data and MEGANv2.1-based wilting points (Chen and Dudhia, 2001). These default wilting points are likely too low, failing to capture drought impacts like those of 2011 and 2012 (Jiang et al., 2018; Sindelarova et al., 2014). Besides wilting point selection, soil moisture inputs, which are challenging to simulate globally, further affect the performance of drought stress algorithms (Opacka et al., 2022; Potosnak et al., 2014b; Wang et al., 2022). Jiang et al. (2018) developed the MEGAN3 approach of using a photosynthesis parameter and a soil wetness factor to determine the drought activity factor, which improves the simulation of isoprene emissions in non-drought and drought periods at the Missouri Ozark AmeriFlux (MOFLUX) field site. Klovenski et al. (2022) found that emulating MEGAN3's drought stress representation in GISS-ModelE, a current development strand of GISS-E2.1-G, requires model-specific tuning and that applying drought stress factors can improve simulations for models that overestimate isoprene emissions. Other drought indicators (e.g. the ratio of actual evapotranspiration to potential evapotranspiration) may be more effective than soil moisture in capturing drought impacts (Wang et al., 2022). These global simulations underpin the importance of accurately simulating the drought-induced response of isoprene emission in ESMs, as well as the need to expand studies beyond MOFLUX to diverse ecosystems to validate and refine drought stress algorithms for ESMs.

### 4.1.3 Uncertainty in modelling $CO_2$ effects on isoprene emissions

Changes in atmospheric $CO_2$ levels are expected to greatly alter biogenic isoprene emissions. With $CO_2$ levels predicted to be doubled under the SSP3-7.0 scenario (and even higher increases under SSP5-8.5 or smaller increases under SSP2-4.5 and SSP1-2.6) by the end of this century, producing accurate models of the $CO_2$ effects on isoprene emissions is fundamentally important. In fact, $CO_2$ effects, including the fertilization effect, increase vegetation productivity (LAI/GPP), which is a key driver of photosynthesis and isoprene emission. However, considerable inter-model differences were found for simulation of this effect within the C4MIP project (Friedlingstein et al., 2006; Hajima et al., 2014; Arora et al., 2020). Models overestimating the LAI/GPP response to rising $CO_2$ are likely to overestimate $CO_2$ fertilization effects on ecosystem productivity and ensuing isoprene emission uncertainties. Direct $CO_2$ inhibition effects also lack clear consensus among models. Understanding the physiological response of isoprene emissions to the combined effects of rising $CO_2$ and temperature is vital for projecting future emissions under climate change. However, the mechanisms controlling isoprene emissions remain uncertain, yielding conflicting hypotheses. Some results of studies suggest that future high $CO_2$ levels might suppress isoprene emission, potentially counteracting the stimulating effects of rising temperatures or the combined effects of temperature increase and enhanced vegetation productivity, which might result in minimal change compared to the present day (Arneth et al., 2007b; Pacifico et al., 2012), an overall increase (Heald et al., 2009), or even a decrease in global isoprene emissions (Hantson et al., 2017). Alternatively, others propose that $CO_2$ inhibition might become less effective at high temperatures (above 30 °C) (Lantz et al., 2019; Monson et al., 2016; Potosnak et al., 2014a; Sun et al., 2013), potentially leading to a net increase in emissions driven by temperature (Lantz et al., 2019; Potosnak et al., 2014a). In contrast, some studies imply stable emission under preindustrial conditions with lower $CO_2$ levels (Heald et al., 2009); others suggest higher emissions than those of recent times (Arneth et al., 2007a; Hantson et al., 2017; Lathière et al., 2010; Pacifico et al., 2012; Unger, 2013). Both sets of modelling studies entail uncertainties; yet, in the absence of measurements or proxies, modelling serves as our sole resort for estimating preindustrial isoprene emission rates.

From preindustrial times to the present day, some models have exhibited a decline in isoprene caused by strong $CO_2$ inhibition (Arneth et al., 2007a; Lathière et al., 2010; Pacifico et al., 2012), similarly to UKESM1-0-LL(P2011) in this study. Others imply no trend with weak inhibition (Heald et al., 2009), similarly to CESM2-WACCM(G2012) and NorESM2-LM(G2012) in this study. However, CMIP6 and VISIT-S3(G1997) show consistency in key climate variables (temperature, radiation, and precipitation) on a global anomaly basis (Fig. S5). These models share the same photosynthesis schemes, explaining the identical upward trends found for GPP/LAI represented for the $CO_2$ fertilization effect (Fig. S6). Therefore, the discrepancy in $CO_2$-driven isoprene emission trends likely derives from differences in the implemented $CO_2$ inhibition schemes.

Both the MEGAN and P2011 models base their $CO_2$ inhibition scheme on the same observational experiment of two mild temperate vascular plants (Possell et al., 2005). However, as presented in Fig. 1 of Heald et al. (2009) and Fig. 8 of Arneth et al. (2007b), the two models diverge considerably in their response curves. Specifically, the MEGAN-based scheme would only reduce isoprene emissions for $CO_2$ levels above 365 ppm and at a lower rate than the P2011-based scheme. Both models normalize emission rates based on the values for the year 2000 (Arneth et al., 2007b; Heald et al., 2009; Possell et al., 2005), but there is a difference in the $CO_2$ inhibition factor for the preindustrial period.

At 280 ppm of $CO_2$ (preindustrial level), the MEGAN-based scheme maintains a $CO_2$ inhibition factor of around 1, whereas the P2011-based scheme gives a $CO_2$ inhibition fac-

tor of approximately 1.5 (Arneth et al., 2007b). This higher value found for the latter implies a 30 % increase in isoprene emissions efficiency in the preindustrial period and a subsequent decrease to present-day levels. In fact, this difference likely explains the divergent long-term emission trends presented in Fig. 4.

Additional uncertainty arises at lower $CO_2$ concentrations (e.g. 185 ppm of the Last Glacial Maximum), for which MEGAN implies stability (Heald et al., 2009) and P2011 estimates much higher emissions than those of the present day (Hopcroft et al., 2017; Pacifico et al., 2012). This discrepancy might add further uncertainty to the interpretation of trends and their influence on atmospheric composition (i.e. methane lifetime) in past atmospheres (Achakulwisut et al., 2015; Hopcroft et al., 2017). Although several studies have consistently demonstrated that elevated $CO_2$ levels inhibit isoprene emissions (Feng et al., 2019; Niinemets et al., 2021; Possell et al., 2005; Possell and Hewitt, 2011), only one study has demonstrated a marked increase at low $CO_2$ levels (185 ppm) (Possell et al., 2005). Further research is necessary to refine $CO_2$ inhibition–enhancement parameterizations and to enhance our understanding of this complex relation at low $CO_2$ levels. Additionally, some results of short-term field studies show that high temperatures might weaken $CO_2$ inhibition, necessitating its integration into models for accurate future predictions (Lantz et al., 2019).

Furthermore, a recent study (Niinemets et al., 2021) has highlighted the diversity in plant responses to $CO_2$-induced isoprene emission changes. Some species such as poplar (common in temperate and boreal regions) are apparently more sensitive than others, such as oak and mango (found in temperate to tropical regions). The variation in plant $CO_2$ responsiveness can be attributed to differences in substrate availability, implying that using a single $CO_2$ inhibition function and threshold for all plants in emission models might overestimate or underestimate this effect. Additionally, the existing scheme relies primarily on temperate species, raising concerns about its accuracy for tropical species (Pacifico et al., 2012; Young et al., 2009). Because tropical plants are the major contributors to global isoprene emissions, better understanding of their long-term response to $CO_2$ is valuable. Developing diverse $CO_2$ inhibition functions for different species, especially those in tropical regions, is necessary for accurate prediction of the effects of rising $CO_2$ on future isoprene emissions and, in turn, their effects on Earth's atmosphere and ecosystems.

## 4.2 Suggestions for future development

As explained above, the reliance on single-function $CO_2$ inhibition schemes for all plant species hinders accurate predictions. Future models must incorporate diverse PFT-specific inhibition functions, particularly for tropical species, which dominate global isoprene emissions. Furthermore, long-term studies across ecosystems must be conducted to confirm the weakening effects of high temperatures on $CO_2$ inhibition for a diversity of plant species. Then this effect must be integrated into current models. Accurate representation of LAI and GPP in land models is extremely useful for simulating the effects of $CO_2$ fertilization on isoprene emissions. Validation against high-resolution satellite imagery, especially in tropical regions with complex vegetation cover, is necessary for this purpose. Current uncertainties related to LULCC effects might also arise from the employment of constant emission factors for PFTs. For instance, deforestation might decrease emissions by replacing high-emitting broadleaf trees with crops, but oil palm, a higher emitter than broadleaf trees, can increase emissions in some cases (e.g. Malaysia) (Misztal et al., 2011; Opacka et al., 2021; Stavrakou et al., 2014; Tanaka et al., 2012). Additionally, neglecting grass and shrub fractions beyond tree cover can contribute further to LULCC effect uncertainty.

Even though random forest regression can replicate global isoprene emissions from all CMIP6 models (Fig. S2), we advocate for the additional multi-model intercomparison of land system models under the TRENDY project's GPP estimation protocols (Friedlingstein et al., 2022; Sitch et al., 2015; TRENDY Portal, 2024), applying similar settings to isoprene emissions. Such multi-model intercomparison will help to pinpoint the important contributors to uncertainty in isoprene emission estimates. Furthermore, leveraging and expanding the existing FLUXNET network (Baldocchi et al., 2001; FLUXNET Portal, 2024) ground-based isoprene emissions observation can provide valuable data for validating long-term isoprene emission models across various regions. We can also develop independent data-driven estimates using machine-learning methods based on these long-term data. These estimates can enhance our understanding of historical isoprene emission changes and can improve the reliability of future predictions.

Current ESMs, including the latest CMIP6 experiments, rely primarily on only two schemes for estimating past, present, and future isoprene emissions: photosynthesis-based models (e.g. P2011) and empirically based models (e.g. MEGAN). These approaches offer some benefits, but they entail some limitations. The photosynthesis-based models estimate isoprene emissions based on photosynthetic electron transport products, capturing light dependence. However, evidence suggests this might not accurately reflect long-term responses to temperature and $CO_2$ changes that are invaluable for decadal and millennial emission projections (Sharkey and Monson, 2014). Additionally, these models do not account for potential substrate effects under future climate conditions, which can strongly influence $CO_2$ and temperature dependence of isoprene emissions. The empirical model estimates isoprene emissions based on two factors: the temperature dependence of enzyme–substrate interactions and empirical data of emission reductions observed for plants grown under different atmospheric $CO_2$ concentrations. However, the adjustments to some driving parameters

in MEGAN lack a clear mechanistic connection to underlying biochemical processes (Monson et al., 2007; Sharkey and Monson, 2014). The reliance on these two schemes with their inadequate representation of biochemical processes, can engender inaccuracies and uncertainties in predicting long-term isoprene emissions under varying environmental conditions. To overcome this limitation, more intensive and comprehensive studies must be conducted to develop a broader range of isoprene and other BVOC emission schemes that better capture the complexity and diversity of biogenic emissions. By incorporating a broader array of emission models, researchers can augment the accuracy and reliability of BVOC emission predictions, especially in the context of evolving environmental conditions and climate scenarios. This diversity in BVOC emission schemes is extremely valuable for advancing our understanding of biogenic emissions and their effects on atmospheric chemistry and climate dynamics. Moreover, this diversity underscores the necessity for additional research to refine the representation of BVOC emissions in ESMs.

## 5   Conclusions

This study comprehensively analysed trends in isoprene emissions and their controlling factors during 1850–2014 using long-term isoprene emissions datasets derived from offline simulations of the VISIT dynamic global vegetation model and online estimates from CMIP6 ESMs. The models, except for UKESM1-0-LL(P2011), incorporate empirical schemes such as MEGAN, categorized into four groups based on their isoprene emission schemes: (1) MEGAN with $CO_2$ fertilization only, which is VISIT(G1997); (2) MEGAN with $CO_2$ effects (fertilization and inhibition), which comprise CESM2-WACCM(G2012) and NorESM2-LM(G2012); (3) MEGAN without $CO_2$ effects, which comprise GFDL-ESM4(G2006) and GISS-E2.1-G(G1995); and (4) photosynthesis-based with $CO_2$ effects, which is UKESM1-0-LL(P2011).

In the present day (2000–2014), mean global isoprene emissions estimated from all models are consistent, with an inter-model spread of only 24 TgC yr$^{-1}$ (5 %), ranging from 434 to 510 TgC yr$^{-1}$. However, regional emissions vary considerably, with an inter-model spread ranging between 0.53 and 30.77 TgC yr$^{-1}$ (9 %–212 %), primarily because of differences in PFT composition and emission factors. Standardizing global PFT maps with specific emission factors can reduce these uncertainties and can improve simulation consistency across models.

Over the historical period examined for this study (1850–2014), isoprene emission trends vary widely across models. Empirical models without $CO_2$ effects (GFDL-ESM4(G2006) and GISS-E2.1-G(G1995)) show slightly increasing trends, whereas the model considering only the $CO_2$ fertilization effect, VISIT-S3(G1997), estimates a significant increasing trend. Models including both $CO_2$ effects (fertilization and inhibition) show no change (CESM2-WACCM(G2012) and NorESM2-LM(G2012)). The sole photosynthesis-based model, UKESM1-0-LL(P2011), exhibits a sharply decreasing trend. These variations in global long-term trends are attributable to differences in the main drivers among models. Similarly to VISIT(G1997) with only $CO_2$ fertilization, MEGAN-based models with $CO_2$ effects (CESM2-WACCM(G2012) and NorESM2-LM(G2012)) emphasize $CO_2$ fertilization, potentially underestimating $CO_2$ inhibition. Also, UKESM1-0-LL(P2011) suggests that $CO_2$ inhibition outweighs fertilization, possibly because of its distinct representation of $CO_2$ inhibition. MEGAN-based models without $CO_2$ effects (GFDL-ESM4(G2006) and GISS-E2.1-G(G1995)) attribute the trend primarily to climate factors, chiefly rising temperatures.

Globally, models vary widely in their estimates of $CO_2$ effects on isoprene emissions, in both direction and magnitude, alongside moderate differences in LULCC-induced emission reductions and relative consensus on climate-driven emission increases. Divergence in $CO_2$-driven emission trends likely stems from models' different $CO_2$ inhibition representations, which can counteract increasing isoprene emission trends attributable to rising temperatures or in combination with $CO_2$ fertilization. At the grid cell level, the highest inter-model variability in simulated isoprene emission trends occurs in regions such as the Amazon, Southeast Asia, and southeastern South America, influenced primarily by $CO_2$ and LULCC.

The discrepancies among models highlight the importance of studying isoprene emission trends and the caution which is necessary for interpreting plant–climate interactions using long-term isoprene emission estimates. Results of our study emphasize the need for deeper investigation of $CO_2$ and LULCC effects on isoprene emissions because their influence on long-term trends far surpasses short-term variations induced by climate factors. Expanding long-term observation networks and refining models by considering diverse species-specific responses to changing $CO_2$ levels in different ecosystems are necessary. Current-generation ESMs rely on empirical and photosynthesis-based approaches to estimate isoprene emissions, each with their idiosyncratic benefits and limitations. Developing more comprehensive emission schemes that better reflect the complexity of plant emissions would support more accurate and reliable predictions of how these emissions can be expected to change under different climate conditions, which is necessary for understanding plant–climate interactions via emissions.

*Code availability.* The VISIT model source code is available at https://doi.org/10.5281/zenodo.13883464 (Do et al., 2024a). The source code used to reproduce the analyses, plots, and tables of this work is archived at http://doi.org/10.5281/zenodo.12754163 (Do et al., 2024b).TS5

*Data availability.* The VISIT simulations are produced using the VISIT model source code. CMIP6 simulations are archived at the Earth System Grid Federation. They are freely available to download. The data of ESGF are accessible via the website interface https://esgf-node.llnl.gov/search/cmip6/ (ESGF, 2023). Both the VISIT and the CMIP6 datasets can also be obtained from the repository at http://doi.org/10.5281/zenodo.12754163 (Do et al., 2024b). All the relevant references to the data used are provided in Table 2, as stated in the text.

*Supplement.* The supplement related to this article is available online at [the link will be implemented upon publication].

*Author contributions.* NTND conducted VISIT simulations, analysed and interpreted trends and discrepancies in CMIP6 isoprene emission data compared to VISIT simulations, and drafted the main text of the paper. KS conceived of the research idea and supervised the findings and paper preparation. AI developed the VISIT model code; pre-processed the land cover, land use, and meteorological data for the simulations; and discussed the results. LKE, VN, KT, ØS, GAF, and DIK contributed greatly to conducting the CMIP6 simulations, data preparation, interpretation, and writing of the manuscript.

*Competing interests.* The contact author has declared that none of the authors has any competing interests.

ther geographical representation in this paper. While Copernicus Publications makes every effort to include appropriate place names, the final responsibility lies with the authors.

*Acknowledgements.* We gratefully acknowledge the Interdisciplinary Frontier Next Generation Researcher Program of the Tokai National Higher Education and Research System for their invaluable support. We also express our deep appreciation to the National Institute for Environmental Studies (NIES) in Japan for providing access to the powerful NEC SX-Aurora TSUBASA supercomputer, which has enabled the VISIT simulations. Ngoc Thi Nhu Do acknowledges the support provided by a Japanese Ministry of Education, Culture, Sports, Science and Technology (MEXT) scholarship. Our sincere thanks are also extended to climate modelling groups for their contributions in generating and sharing their model outputs, to the Earth System Grid Federation (ESGF) for providing efficient data storage and access, and to the many funding agencies which have made CMIP6 and the ESGF possible. Also, Gerd A. Folberth wishes to acknowledge support by the Met Office Hadley Centre Climate Programme, funded by DSIT with additional funding through the EU Horizon project ESM2025 (grant 101003536). Louisa K. Emmons acknowledges support from the NSF National Center for Atmospheric Research, which is a major facility sponsored by the US National Science Foundation under cooperative agreement no. 1852977.

*Financial support.* The research has been supported by the Ministry of the Environment, Government of Japan (Global Environment Research Fund (grant no. S-20)) and by the Japan Society for the Promotion of Science (KAKENHI (grant nos. 20H04320 and 23H04971)). TS6

*Review statement.* This paper was edited by Patrick Jöckel and reviewed by two anonymous referees.

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

**Remarks from the language copy-editor**

CE1     In keeping with the dominant style of the manuscript, Oxford spellings (-ize endings, e.g. "characterize") were not removed.

CE2     Please confirm this unit is correct throughout (with "$\mathrm{yr}^{-2}$" rather than "$\mathrm{yr}^{-1}$").

CE3     Please confirm the word choice (rather than "Currently").

CE4     Please decide whether any of the abbreviations in this column need further definition for clarity.

CE5     Please check that slashes are used to mean "or" throughout the paper. If you meant "and" or "and/or" in some or all instances, the slash should be replaced accordingly.

CE6     Dashes have been formatted to denote minus signs throughout this subsection. Please check this is correct.

CE7     Please check the placement of this reference is correct following edits to the sentence.

CE8     Please note edits to this table carefully because changes may not appear in the track-changes file.

CE9     Please confirm the word change here is correct or clarify the meaning.

**Remarks from the typesetter**

TS1     Please note that the first names will be abbreviated for the citation of the article; please confirm "Do, N. T. N.".

TS2     Please confirm running title or provide an alternative.

TS3     Please confirm.

TS4     Not mentioned in the reference list.

TS5     You have additionally added http://doi.org/10.5281/zenodo.12754163 as an interactive computing environment (ICE) to our system. Please confirm that your interactive computing environment provides a well-defined structure for data, code, text, documentation, runtime environment, and user interface controls for a piece of research. An interactive computing environment should be complete so that everything one needs to execute is available and no other assets must be downloaded. If this is not the case, we will insert it as code.

TS6     Please note that there is a discrepancy between funding information provided by you in the acknowledgements and the funding information you indicated during manuscript registration, which we used to create this section. Please double-check your acknowledgements to see whether repeated information can be removed from the acknowledgements or changed accordingly. If further funders should be added to this section, please provide the funder names and the grant numbers. Thanks.

TS7     Please ensure that any data sets and software codes used in this work are properly cited in the text and included in this reference list. Thereby, please keep our reference style in mind, including creators, titles, publisher/repository, persistent identifier, and publication year. Regarding the publisher/repository, please add "[data set]" or "[code]" to the entry (e.g. Zenodo [code]).

TS8     If possible, please provide a title.

TS9     If possible, please provide a title or authors. Another possibility would be to move the URL directly into the text.

TS10     Please provide volume.

TS11     Please provide the publisher.

TS12     Please provide editors (if not authors), the publisher and a persistent identifier.

TS13     Please provide volume.

TS14     Please provide editors (if not authors) and a persistent identifier.

TS15     Please provide volume.

TS16     Please provide all author names or let me know if this should be cited as "IPCC (2012)".

TS17     If possible, please provide a title or authors. Another possibility would be to move the URL directly into the text.

TS18     Please provide journal, volume, and page range or DOI and article number.

TS19     Please provide editors (if not authors), the publisher and a persistent identifier.