# Peer review of "Historical Trends and Controlling Factors of Isoprene Emissions in CMIP6 Earth System Models"

_EGUsphere, 2024_

## Author Response (AR1)

Author's response to the Anonymous Referees for the manuscript titled "**Historical Trends and Controlling Factors of Isoprene Emissions in CMIP6 Earth System Models**" by Thi Nhu Ngoc Do and co-authors.

We would like to thank the referees for the helpful comments. We have revised the manuscript in response to their comments and hope it is now suitable for publication. Below, the referees' comments are presented in *blue italics*, with our responses in regular font. The revisions made to the manuscript are highlighted in red, and the corresponding line numbers show where each change has been made in the revised version.

**Reply to Referee #1**

*This paper by Do et al. examines the differences in historical trends and controlling factors of isoprene emissions across six modelling approaches and highlights the importance of better parameterizing the CO2 effect, as well as the global/regional PFT distribution and LULCC effects, on isoprene emissions. The study indicates that the influence of CO2 and LULCC effects on long-term isoprene emissions surpasses climate factors, which is currently not well understood. As the increase of CO2, LULCC and temperature are happening together, I feel this work comparing these controlling factors is quite important.*

*Overall, I found the quality of the work strong, with appropriate and justified methods. Hence, the conclusions are strongly supported by the evidence provided. The suggestions I make below are primarily to increase clarity and conciseness but in themselves are minor things that don't significantly question the findings of this work.*

Thank you for the encouraging comments. We have made every effort to address your helpful comments to improve clarity and conciseness in the revised manuscript. Please find detailed responses to your comments below.

*Lines 17-18: The influence of isoprene on the Earth's radiative balance, as you very nicely described in the introduction (lines 51-56). Therefore, I suggest you rewrite this sentence by clarifying the indirect effect of isoprene on the radiative balance.*

Thank you for your helpful comments. We have revised the sentence to better clarify the indirect effect of isoprene on the Earth's radiative balance. The revised sentences (Lines 17–18) are as follows:

"Terrestrial isoprene, a biogenic volatile organic compound emitted by many plants, indirectly influences the Earth's radiative balance through its interactions with atmospheric oxidants, affecting ozone formation, methane lifetime, and secondary aerosol production."

*Lines 375-376: From a visual analysis, I would say that VISIT-S3(G1997) and UKESM1-0-LL(P2011) show higher emissions in the northwestern Amazon, which is interesting and in line with satellite retrievals of isoprene concentrations over this region (doi:10.1029/2021JD036181). I suggest mentioning/citing this here.*

Thank you for your helpful comment. We have revised the text to mention the higher emissions in the northwestern Amazon as simulated by VISIT-S3(G1997) and UKESM1-0-LL(P2011), which align with satellite retrievals of isoprene concentrations in this region. The revised sentences (Lines 377–380) are as follows:

"VISIT-S3(G1997) and UKESM1-0-LL(P2011) also identify the Central Amazon as the emission hotspot but show relatively higher emissions in the northwestern Amazon. This pattern aligns with satellite retrievals of isoprene concentrations over this region (Wells et al., 2022), though the emissions are of a smaller magnitude compared to those simulated by CESM2-WACCM(G2012) and NorESM2-LM(G2012)."

*Lines 376-377: I do not see a hot spot of isoprene emission in the northern Amazon for the simulations with GFDL-ESM4(G2006). It seems there is a hot spot in the central and western Amazon, but it is a weak sign compared to other simulations. Was it just a visual analysis?*

Thank you for your helpful comment. We have revised the text to correct the identification of the western Amazon as an emission hotspot in the GFDL-ESM4 (G2006) model and have incorporated your clarification regarding the weaker signals in the revised sentences (Lines 380–382) below:

"However, GFDL-ESM4(G2006) and GISS-E2.1-G(G1995) models respectively recognize the western Amazon and northern Amazon as emission hot spots, respectively, although this signal is weaker compared to other models".

This result was based on a visual analysis of the overall hotspot identification for emissions. However, the figures showing emissions across the entire Amazon can be found in Table S4. This table shows that emissions are highest in the CESM2-WACCM(G2012) and NorESM2-LM(G2012) models, moderate in VISIT-S3(G1995) and UKESM1-0-LL(P2011), and lowest in GFDL-ESM4(G2006) and GISS-E2.1-G(G1995). This distribution aligns with the spatial emission patterns observed across the models.

*Lines 537-540: I found this confusing. Wasn't the precipitation dataset the same for both models? Can you please explain why you have increased precipitation and reduced precipitation in the same Amazon region? Maybe I missed this point, but I think it is worth explaining it here.*

Thank you for your helpful comment. We apologize for the confusion. In response, we have clarified in the manuscript that the climate datasets (including precipitation) differ between the models. The

input datasets for VISIT(G1997) and the Random Forest emulator applied to each CMIP6 model are described in Sections 2.1.2 (Lines 202–203) and 2.2.2 (e.g., Lines 264–270), respectively. We hope this revision addresses your concern. The revised sentences (Lines 549–553) are as follows:

"Precipitation-driven changes remain highly uncertain, both in magnitude and sign of the trends. For example, VISIT(G1997) shows marked increases in Amazonia emissions because of increased precipitation, whereas UKESM1-0-LL(P2011) projects a decrease in the same region because of reduced precipitation. This discrepancy results from the different climate datasets: VISIT(G1997) relies on reanalysis data, while UKESM1-0-LL(P2011), like other CMIP6 models, uses its own modelled climate data (Fig. S7)."

*Line 678: it is not usual to present a new figure/result in the discussion section. I suggest moving figure 14 to the results.*

Thank you very much for your comment. In the revised manuscript, we have moved Figure 14 to the Results section and revised the text accordingly. For your reference, please see lines 583–599.

*Lines 619-691: In general, this whole section seems more like results (some repetition from the results) than discussion. I suggest integrating it to the results or reducing it.*

Thank you very much for your suggestion. We have integrated the results of the inter-model spreads of each factor's contribution to isoprene emission trends (Fig. 14) into the Results section. The revised manuscript now focuses on discussing the variability in the attribution of isoprene emission trends among the models. As a result, we have adjusted the content in Lines 619–691 (now Lines 663–730) to better align with the Discussion section, ensuring that it complements the Results without redundancy.

*Line 714: Do you mean "both sets of modelling studies"? You have cited a few studies in the previous sentence. I suggest you rewrite this sentence for clarification.*

Thank you for your helpful comment. We have revised the sentence for clarity. In the revised manuscript, we now refer to "both sets of modelling studies" as your suggestion to better distinguish the different study groups. The revised sentences (Lines 751–752) are as follows:

"Both sets of modelling studies entail uncertainties; yet, in the absence of measurements or proxies, modelling serves as our sole resort for estimating preindustrial isoprene emission rates."

*Reply to Referee #2*

*The paper egusphere-2024-2313 presents a well-designed study to evaluate historical isoprene emission trends and their controlling factors from online simulations of CMIP6 Earth System Models and one offline simulation using the VISIT dynamic vegetation model. The endeavor of this work to elucidate the historical changes of global isoprene emissions during 1850-2014 due to climate, land cover changes and $CO_2$ concentrations is deeply appreciated. The concurrent use of VISIT facilitates efficient exploration of isoprene emission sensitivity to various driving factors. Specifically, VISIT offers a way to compare the $CO_2$ effects on isoprene emissions that are incorporated in three of the CMIP6 models, since VISIT includes the fertilization effect of $CO_2$, but not the direct inhibition by $CO_2$. The main result of the model comparison is that $CO_2$ concentrations and land cover changes are the primary drivers for isoprene emission trends in CMIP6 models that include $CO_2$ effects. This is an important finding and emphasizes the need for accurate representation of land cover changes and $CO_2$ effects in Earth System models. The paper is organized in a clear fashion and the presentation quality is excellent. The discussion of model uncertainties is quite comprehensive and future directions are in accordance with the findings of this work. However, the discussion should be further expanded to address the links to plant phenological models, which could support the understanding of historical and future changes in plant species composition and their emission behavior due to climate changes.*

Thank you for the encouraging comments. We have made every effort to address your helpful comments to improve clarity and conciseness in the revised manuscript. Please find detailed responses to your specific comments below.

*Specific Comments:*

*1.) Evaluation of the model sensitivities to climate variables, i.e. temperature, shortwave radiation, and precipitation, would benefit from supplementing their global and probably also regional trends during 1850-2014. Global trends of temperature for each model are provided in Table 3. At least, the global trends of shortwave radiation and precipitation should be provided in tabular form. It is also suggested to present their regional trends for regions with highest isoprene emissions or on a global map. This inclusion would also facilitate the interpretation of Figure 12. Obviously, precipitation trends are not consistent among models. On page 25, lines 538-540, it is stated that different precipitation trends across the models gave rise to different isoprene emission trends in Amazonia.*

Thank you for your thoughtful suggestion. We have added global maps in Figure S7 showing trends of climate variables (temperature, shortwave radiation, and precipitation) for the 1850–2014 period to aid in the interpretation of Figure 12. Global trends for these variables are also provided in Table

S5. Additionally, we have revised the sentence on lines 538–540 (now lines 550–553) to clarify the impact of varying precipitation trends across models on isoprene emissions. The revised sentence is as follows:

"For example, VISIT(G1997) shows marked increases in Amazonia emissions because of increased precipitation, whereas UKESM1-0-LL(P2011) projects a decrease in the same region because of reduced precipitation. This discrepancy results from the different climate datasets: VISIT(G1997) relies on reanalysis data, while UKESM1-0-LL(P2011), like other CMIP6 models, uses its own modelled climate data (Fig. S7)."

*2.) Figure 6 and 8 shows for VISIT(G1997) the absence of interannual variability in the response of isoprene emissions to the drivers of land cover change and climate during 1850-1900, while the CMIP6 models display variability also during this historical period. Is this due to different land cover and the climate reanalysis data used in VISIT or are there any compensating mechanisms at work that balance the emission response? The spin-up phase should have been sufficiently long. Given the high sensitivity of isoprene emissions to temperature changes (Figure 13), the missing variability in VISIT during 1850-1900 looks like an artifact. On page, lines 478-480: "VISIT(G1997) notably exhibits lower interannual variation than the CMIP6 models." This discrepancy should be explained by analyzing the causality for the weak response to interannual changes in climate variables.*

Thank you very much for your comment. We have updated lines 478–480 (now lines 483–493) to explain the lower interannual variation of isoprene emissions in VISIT compared to the CMIP6 models. The revised sentences are as follows:

"Empirical-based CMIP6 models agree that temperature is the primary driver of interannual variation of isoprene emissions, whereas the photosynthesis-based model UKESM1-0-LL(P2011) highlights radiation and precipitation, and VISIT(G1997) emphasizes temperature and precipitation over radiation (Fig. 8). Interannual variations in these climate variables primarily drive interannual changes in isoprene emissions across models. VISIT(G1997) notably exhibits lower interannual variation in isoprene emissions than that in the CMIP6 models, primarily because it relies on reanalysis climate data, which contains less variability compared to the climate data simulated by the CMIP6 models (Fig. S5 and Table S5). The small interannual variability during 1850–1900 simulated by VISIT could come from the spin-up method using climatological forcing data, with the model being repeatedly driven by the climate data of 1901 during this period. Furthermore, although GISS-E2.1-G(G1995) uses a temperature-dependence algorithm similar to that of VISIT(G1997) (Table S1), the greater temperature variability in GISS-E2.1-G(G1995) results in larger interannual variation in isoprene emissions."

As described in Section 2.1.2, each simulation included a spin-up phase lasting 300 to 3000 years, depending on the biome type for each grid cell. This phase was initialized with $CO_2$ levels from 1700, 1901 climate data, and constant land use from 1700. It was followed by two transient periods: 1700–1900 and 1901–2021. During the 1850–1900 period of the transient 1700–1900 phase, VISIT(G1997) shows no interannual variation in Figures 6 and 8 because the climate data for 1901 was repeatedly used, as reanalysis data begins in 1901. The VISIT(G1997) model outputs were extracted for 1850–2014 to align with the CMIP6 historical simulations. Although the lack of interannual variation for 1850–1900 is acknowledged, it has minimal impact on the long-term trends in isoprene emissions.

*3.) Precipitation is found to play a minor role in controlling long-term trends of isoprene emissions in most CMIP6 models. The effect of extreme climate, such as drought stress, on emissions of isoprene from the vegetation is briefly mentioned in the Introduction, but not further explored in this study. Jiang et al. (2018) developed the MEGAN3 approach of using a photosynthesis parameter and a soil wetness factor to determine the drought activity factor, which improves the simulation of isoprene emissions in non-drought and drought periods at the Missouri Ozarks AmeriFlux (MOFLUX) field site. Their global simulation underpins the importance to simulate the drought-induced response of isoprene emission accurately in Earth System models. MEGAN v2.1 considers the effects of drought using soil moisture and wilting point. Different wilting point data in MEGAN can lead to substantially different outcomes for the effect of drought. The wilting points from Chen and Dudhia (2001) that is default for MEGAN v2.1 seem to be too low, and consequently the model does not capture the 2011 or 2012 drought effect on isoprene emissions (e.g., Sindelarova et al. 2014).*

Thank you very much for your comment. We agree that precipitation plays a minor role in controlling long-term isoprene emission trends in most CMIP6 models. While this study focuses on long-term trends, we appreciate your suggestion to consider incorporating drought effects on isoprene emissions in future ESM development. We have added this discussion to Lines 714–730 and cited the references you recommended (Jiang et al. 2018; Chen and Dudhia, 2001; Sindelarova et al., 2014). Please kindly refer it for your reference.

*4.) Changes in tree species distribution and composition in response to climate change impact the amount and composition of BVOC emissions. The isoprene emission rate varies significantly across plant species. The assignment of emission factors to certain PFT is often not unambiguous. In this regard, the use of plant-specific emission factors is expected to better reflect the impact due to changes of individual climate drivers. Notably, emission factors (emission potentials) can vary significantly even within the same genus (Karl et al., 2009; Satake et al., 2024). Dani et al. (2014) suggested that the trait of isoprenoid emission in evergreen plants can be lost during evolution in*

*favor of more storable compounds (monoterpenes) to better cope with repeated and prolonged stress. The proportion of isoprene-emitting tropical trees appears to increase with mean annual temperature but to decrease with length of dry season (Taylor et al., 2018). The discussion in section 4.1.1 should be extended to emphasize the aspects of species composition changes and plant phenological changes due to changing climate, citing the above-mentioned literature.*

Thank you very much for your comment. We have revised Section 4.1.1 (Lines 651–661) to include a discussion on how changes in species composition and plant phenology influence isoprene emissions in response to climate change, citing your suggested references (Karl et al., 2009; Dani et al., 2014; Taylor et al., 2018; Satake et al., 2024). This revision emphasizes the importance of using plant-specific emission factors rather than generalized PFT-emission factors to improve predictions. We hope this revision addresses your concerns.

*Technical Corrections:*

*Table 1: there are missing entries for S2 and S3.*

Thank you for your comment. We have added the "-" to indicate variables that varied annually for temperature and precipitation in S2 and S3 in Table 1.

*Figure 6: it would be beneficial for the reader to insert a thin black horizontal line at zero.*

Thank you for your thoughtful suggestion. Following your recommendation, we have added a thin black horizontal line at zero in Figure 6 (and Figure 8) to enhance clarity for the reader.

*References:*

*Chen, F., and Dudhia, J.: Coupling an advanced land surface-hydrology model with the Penn State–NCAR MM5 modeling system. Part I: model implementation and sensitivity, Mon. Wea. Rev., 129, 569–585, 2001.*

*Dani, K. G. S., Jamie, I. M., Prentice, I. C., Atwell, B. J.: Evolution of isoprene emission capacity in plants, Trends Plant Sci., 19(7), 439–446, doi:10.1016/j.tplants.2014.01.009, 2014.*

*Jiang, X., Guenther, A., Potosnak, M., Geron, C., Seco, R., Karl, T., Kim, S., Gu, L., and Pallardy, S.: Isoprene emission response to drought and the impact on global atmospheric chemistry, Atmos. Environ., 183, 69–83, doi:10.1016/j.atmosenv.2018.01.026, 2018.*

*Karl, M., Guenther, A., Köble, R., Leip, A., and Seufert, G.: A new European plant-specific emission inventory of biogenic volatile organic compounds for use in atmospheric transport models, Biogeosciences, 6, 1059–1087, doi:10.5194/bg-6-1059-2009, 2009.*

*Satake, A., Hagiwara, T., Nagano, A. J., Yamaguchi, N., Sekimoto, K., Shiojiri, K., and Sudo, K.: Plant molecular phenology and climate feedbacks mediated by BVOCs, Annu. Rev. Plant Biol. 2024. 75, 605–627, https:// doi.org/10.1146/annurev-arplant-060223- 032108, 2024.*

*Taylor, T. C., McMahon, S. M., Smith, M. N., Boyle, B., Violle, C., et al.: Isoprene emission structures tropical tree biogeography and community assembly responses to climate, New Phytol., 220, 435– 446, doi:10.1111/nph.15304, 2018.*
* * *
Thank you once again for your time and consideration.

Kind regards,

Ngoc Do

(On behalf of the co-authors)